# Development of a fixed module repertoire for the analysis and interpretation of blood transcriptome data

Matthew C. Altman [1,2,21✉], Darawan Rinchai [3,21✉], Nicole Baldwin [4], Mohammed Toufiq [3], Elizabeth Whalen[1], Mathieu Garand[3], Basirudeen Syed Ahamed Kabeer[3], Mohamed Alfaki[3], Scott R. Presnell [1], Prasong Khaenam[1], Aaron Ayllón-Benítez [5], Fleur Mougin[5], Patricia Thébault[6], Laurent Chiche[7], Noemie Jourde-Chiche[8], J. Theodore Phillips[4], Goran Klintmalm[4], Anne O'Garra [9,10], Matthew Berry[11], Chloe Bloom[10], Robert J. Wilkinson[12,13,14], Christine M. Graham[9], Marc Lipman[15], Ganjana Lertmemongkolchai [16], Davide Bedognetti[3], Rodolphe Thiebaut [5], Farrah Kheradmand [17], Asuncion Mejias [18], Octavio Ramilo [18], Karolina Palucka[4,19], Virginia Pascual [4,20], Jacques Banchereau [4,19] & Damien Chaussabel [1,3✉]

As the capacity for generating large-scale molecular profiling data continues to grow, the ability to extract meaningful biological knowledge from it remains a limitation. Here, we describe the development of a new fixed repertoire of transcriptional modules, BloodGen3, that is designed to serve as a stable reusable framework for the analysis and interpretation of blood transcriptome data. The construction of this repertoire is based on co-clustering patterns observed across sixteen immunological and physiological states encompassing 985 blood transcriptome profiles. Interpretation is supported by customized resources, including module-level analysis workflows, fingerprint grid plot visualizations, interactive web applications and an extensive annotation framework comprising functional profiling reports and reference transcriptional profiles. Taken together, this well-characterized and well-supported transcriptional module repertoire can be employed for the interpretation and benchmarking of blood transcriptome profiles within and across patient cohorts. Blood transcriptome fingerprints for the 16 reference cohorts can be accessed interactively via: https://drinchai.shinyapps.io/BloodGen3Module/.

A full list of author affiliations appears at the end of the paper.

Technological advances over the past two decades paired with improvements in cost-effectiveness have enabled the large-scale implementation of high throughput molecular profiling approaches. In translational settings, these advances now permit almost routine measurement of molecular phenotypes at very high resolutions, via whole genome, proteome, metabolome, microbiome, and transcriptome profiling[1]. Among those approaches, blood transcriptome profiling has proven especially well-suited for the unbiased assessment and monitoring of immunological responses in patient studies[2,3]. It consists of measuring abundance of transcripts in bulk blood samples or in Peripheral Blood Mononuclear Cell (PBMC) fractions. It employs robust sampling protocols, which are amenable to implementation on large scales, both inside and outside of clinical settings[4,5]. Over the years the approach has gradually become more cost-effective, with the price point for recently introduced 3′ biased counting applications via RNA-seq currently standing at <$100 per sample[6,7]. Blood transcriptome profiling approaches have been employed across virtually all fields of medicine[2]. One of the most common uses has been for the definition of "disease signatures" and investigation of mechanisms associated with, and potentially implicated in, disease pathogenesis[3,8,9]. On larger scales, blood transcriptomics profiling has also served as a basis for the development of biomarkers, for instance in order to improve management and diagnosis of sepsis[10–12]. It has also proven valuable as an immunomonitoring tool in clinical studies, to assess response to vaccines or immune modifying therapies[13–15].

Our group has previously developed fixed repertoires of transcriptional modules which have been employed to support the analysis and interpretation of blood transcriptome data[16–18]. Notably, we have developed such repertoires as reusable analytic frameworks—i.e. with the intent of employing these pre-established and well-characterized module sets for the analysis of newly generated transcriptome datasets. Consequently, our team has released only two different module repertoires over a 12-year period, which we, and others, have used to analyze numerous blood transcriptome datasets [e.g.[19–22]]. With the construction of a new repertoire ("BloodGen3"), we aimed first to increase the range of immunological states upon which the definition of modules would be based, which was limited to seven in our earlier attempt, and second, to improve the resources that would support downstream data analysis and interpretation. Specifically, 16 input datasets were employed for the construction of the BloodGen3 repertoire, comprising 985 unique blood

transcriptome profiles from: patients with autoimmune, infectious, or inflammatory diseases; cancer patients; liver transplant recipients; and pregnant women. Secondly, we developed a comprehensive bioinformatics ecosystem specifically adapted to the BloodGen3 repertoire, to support downstream analysis, visualization, and interpretation of blood transcriptome profiling data. The custom resources that have been developed include an R package which permit to run analysis workflows for group comparison and individual molecular fingerprinting and to visualize module-level data as custom module fingerprint grids and heatmaps. In addition, extensive functional annotations and reference transcriptome profiles for each of the 382 modules comprising the repertoire have been made available via interactive circle packing plots. Finally, web applications have been deployed that give users the ability to dynamically generate fingerprint plots for different collections of reference datasets.

## Results

**Generation of a collection of datasets covering a wide range of immune states.** The development of transcriptional module repertoires relies on identifying gene co-expression events using transcriptome profiling data as a starting point. For this new blood transcriptome module repertoire, we used 16 datasets (GEO ID: GSE100150) that encompassed 985 individual whole blood transcriptome profiles. Each dataset corresponds to a different pathological or physiological state (Table 1). These datasets were generated from whole blood samples processed in the same facility using Illumina HT12 BeadArrays (details are provided in the Methods section). Similar to our first two repertoires (Supplementary Table 1), we included data from patients (adult and pediatric) with: systemic lupus erythematosus (SLE), systemic onset juvenile idiopathic arthritis (SoJIA), liver transplants, and receiving maintenance immunosuppressive therapy, metastatic melanoma, and infectious diseases [with an expanded range that now includes infections caused by influenza, respiratory syncytial virus (RSV), human immunodeficiency virus (HIV) infections, *Mycobacterium tuberculosis, Staphylococcus aureus*, and *Burkholderia pseudomallei* (which causes melioidosis) and sepsis caused by other bacteria (*Streptococcus pneumoniae, Salmonella spp.*, and *Pseudomonas aeruginosa*)]. We also added six new conditions to our framework: inflammatory conditions of the skin (juvenile dermatomyositis), lung [chronic obstructive pulmonary disease (COPD)] and circulation (Kawasaki disease); multiple sclerosis (MS); primary immune (B-cell) deficiency; and

---

**Table 1 Datasets used for module construction.**

| Dataset | Category | Population | # Samples (Cases) | # Samples (Control) | # Samples (Total) |
|---|---|---|---|---|---|
| 1 *Staphylococcus aureus* infection | Bacterial Infection | Pediatric | 99 | 44 | 143 |
| 2 Sepsis | Bacterial Infection | Adult | 35 | 12 | 47 |
| 3 Tuberculosis | Bacterial Infection | Adult | 23 | 11 | 34 |
| 4 Influenza | Viral Infection | Pediatric | 25 | 14 | 39 |
| 5 RSV | Viral Infection | Pediatric | 70 | 14 | 84 |
| 6 HIV | Viral Infection | Adult | 28 | 35 | 63 |
| 7 Systemic lupus erythematosus | Autoimmune | Pediatric | 55 | 14 | 69 |
| 8 Multiple sclerosis | Autoimmune | Adult | 34 | 22 | 56 |
| 9 Juvenile dermatomyositis | Autoimmune | Pediatric | 40 | 9 | 49 |
| 10 Kawasaki disease | Autoinflammatory | Pediatric | 21 | 23 | 44 |
| 11 Systemic onset juvenile idiopathic arthritis | Autoinflammatory | Pediatric | 62 | 23 | 85 |
| 12 COPD | Inflammatory | Adult | 19 | 24 | 43 |
| 13 Melanoma | Malignancy | Adult | 22 | 5 | 27 |
| 14 Pregnancy | Physiologic variant | Adult | 25 | 20 | 45 |
| 15 Liver transplant recipients | Immunosuppressed | Adult | 94 | 30 | 124 |
| 16 B-cell deficiency | Immunodeficiency | Adult | 20 | 13 | 33 |

Sixteen distinct datasets were used as the input for module repertoire construction. Each dataset corresponds to a different condition or physiological state and comprises both cases and matched controls. Each dataset was processed as a single batch at the same facility with the data generated using Illumina HumanHT-12 v3.0 Gene Expression BeadChips. The collection comprises a total of 985 individual transcriptome profiles.

pregnancy (serving as a physiological variant). Patients with type 1 diabetes or with an *Escherichia coli* infection included in previous repertoires were not included this time.

In summary, the sixteen datasets that were assembled capture a wide range of immunological responses. This should permit the construction of a transcriptional module repertoire that will prove useful as a generic framework for blood transcriptome data analyses.

**Implementation of a stepwise approach to blood transcriptional module repertoire construction.** After collating the 16 input datasets, we next followed a stepwise process to construct the BloodGen3 module repertoire and identify co-expression networks (Fig. 1). We used the module construction algorithm we implemented for the selection of the previous repertoire (the second generation; the code is provided in the Supplementary Material). This approach comprised four main steps: (1) clustering of each individual dataset; (2) recording the number of instances where two genes are included in the same cluster, with the values ranging from 0 to 16 (i.e. reflecting the range of co-clustering in none or all 16 of the datasets); (3) construction of a weighted co-expression network, where the edges between the genes represent at least one co-clustering event in one of the input datasets and the weight is assigned based on the total number of co-clustering events; (Supplementary Fig. 1); and (4) mining the resulting network to identify highly inter-connected sub-networks that form the modules.

The construction of a module repertoire in this manner is thus entirely data-driven and does not rely on any a priori information about gene interactions or functions. In total, we identified 382 modules comprising 14,168 transcripts (95.8% of the transcripts detected in this dataset collection).

**Development of module-level analysis workflows and visualizations.** A key characteristic of the gene sets collected via the process described above is that, by construction, changes in abundance of the corresponding transcripts within a given module will tend to be coordinated. As such, it should be possible to use these modules as a "framework" to: (1) identify functional convergences among the genes that comprise each set, and (2) summarize changes in overall transcript abundance related to pathological processes or therapeutic interventions.

We determined the gene composition of each of the 382 modules comprising the BloodGen3 repertoire (Supplementary Data 1). The average number of genes per module was 37.1, the median was 26.5 and the range was 12–169. Functional profiling and enrichment results were generated using multiple tools (GSAn, Literature Lab, IPA, DAVID, KEGG, BioCarta, OMIM, and GOTERM). We also determined the extent of overlap between the BloodGen3 repertoire and those we obtained earlier (Gen1, Gen2) as well as those constructed by our colleagues at Emory University (BTMs)[23]. For module-level analyses, we determined the proportion of the constitutive transcripts that differ in abundance levels between study groups (e.g. cases vs. controls; pre-treatment vs. post-treatment). Using this approach, we derived two values corresponding to the percent of transcripts that are (i) increased and (ii) decreased. The cut-off points can be chosen based on user preferences. For example, cut-offs can be based on statistics, fold changes and/or differences with or without multiple-testing correction for group comparisons.

Subsequently, the extent of differential expression at the module level can be displayed as a "fingerprint", assigning each module to a fixed position on a grid plot with color-coding according to the level of increased or decreased abundance of the constituent transcripts (Fig. 2). For this, we then performed a

second tier of clustering to group the 382 modules into 38 "aggregates", with each row on the grid displaying the modules corresponding to one such aggregate. Segregation into distinct aggregates was based on similarities in abundance levels observed across the collection of 16 datasets. Using this approach, we derived two levels of granularity (i.e. module level vs. module aggregate level) with the number of variables for interpretation at the least granular module aggregate level constrained to a more manageable number. As a result of this process, the changes in expression levels for each row in the fingerprint grid tend to be coordinated, which was not the case for prior iterations of such fingerprint grids (Fig. 3). Some degree of functional convergence can thus be observed within a given row of modules. As an example, in our fingerprint we found that row A1 comprised several modules associated with lymphocytes, while row A28 comprised six distinct "interferon modules" and rows A33 and A35 comprised a number of modules functionally associated with inflammation.

Overall, fingerprint grid plots can complement traditional heatmap representations. Fixing the positions of modules on the grid allows, with some practice, to identify at a glance immunological/functional characteristics associated with a given blood transcriptome profile.

**Illustrative case of fingerprint grid plot representation.** We next demonstrate the analysis and visualization approach described above with an illustrative case, focusing primarily on SLE, a disease for which the blood transcriptome signature has been well-characterized. Fingerprints of other reference disease cohorts employed for module construction are included to provide additional context.

Data interpretation is facilitated by tiered dimension reduction. The first vertical reading of the fingerprint grid permits visualization of changes across the aggregates, while the horizontal reading permits visualization of changes within an aggregate and across modules. In this illustrative case, we compared the transcriptome profiles among 55 pediatric patients with SLE and 14 healthy control subjects (Fig. 3). We identified an interferon-dominated signature (A28) accompanied by modules associated with cell cycle (A27 and A29, including antibody production). We observed an increase in the abundance of modules associated with inflammation and neutrophils (A35), which is a hallmark of the SLE transcriptome signature. These changes were accompanied by a decrease in transcript abundance, which was more apparent for some modules belonging to aggregates A1, A2, and A3, which are arrayed across the first three rows of the fingerprint grid. More specifically, for the module aggregate A1, the most marked decreases were observed for modules associated with protein synthesis (dark purple color at positions 1, 5, 11, and 19 on row A1).

It is also possible to "aggregate" the changes observed by row, thereby reducing the dimensions for a given dataset even further. In this illustrative case, we reduced the dataset from 382 modules to 27 "aggregates" (Fig. 3a). The decision to take this extra step depends on the desired level of resolution. Mapping changes at the aggregate level will produce the most reduced and simplest fingerprint possible. However, our earlier work also showed that distinct interferon modules are biologically and clinically meaningful[24]. Also when focusing on a given signature or pathway it would be indicated to work at the module level rather than at the aggregate level.

Fingerprint grid plots can be generated for each of the 16 diseases or physiological states via the BloodGen3 web application (https://drinchai.shinyapps.io/dc_gen3_module_analysis/#); seven of them are shown in Fig. 3 as an illustration. In brief, among these fingerprint grid plots, we found that blood transcriptome

**Fig. 1 The module repertoire construction process. a** A collection of 16 blood transcriptome datasets spanning a wide range of immunological and physiological states was used as a starting point for the identification of gene co-expression patterns (RSV: Respiratory Syncytial Virus, HIV: Human Immunodeficiency Virus, COPD: Chronic Obstructive Pulmonary Disease). **b** Each dataset was independently clustered via k-means clustering. **c** Gene co-clustering events were recorded in a table, where the entries indicate in how many datasets, out of the 16, co-clustering was observed for a given gene pair. **d** The co-clustering table served as the input to build a weighted co-clustering graph (see also Supplementary Fig. 1), where the nodes represent genes and the edges represent co-clustering events. **e** The largest, most highly weighted sub-networks among a large network (constituting 15,132 nodes in this case) were identified mathematically and assigned a module ID. The genes constituting this module were removed from the selection pool and the process was repeated to select the next largest set of genes. Once all the gene sets for a given round of selection have been identified the criterion is relaxed for the next round, (e.g. M1 modules corresponding to the first round with the highest co-clustering weight [16 out of 16 datasets], M2 modules corresponding to the second round [co-clustering observed in 15 out of 16 datasets]). Overall, this process resulted in the selection of 382 modules comprising 14,168 transcripts.

perturbations were most widespread in patients with MS and patients with a *Staphylococcus aureus* infection (Fig. 3b), with opposing patterns of change. Changes associated with COPD or stage IV melanoma (Fig. 3c) were very subtle but nonetheless distinct, with differences in transcript abundance compared to control subjects most visible for aggregates concerning oxidative phosphorylation, monocytes, inflammation (A24–A26), erythrocytes and neutrophil activation (A36–A38). We also found differences in the intensities of sets of modules associated with inflammation between these two diseases (A33–A35). Interferon signatures (A28) were a salient trait in patients with SLE (Fig. 3a) and were present in patients infected with viral pathogens (e.g. HIV, in Fig. 3), as well as patients with active tuberculosis (consistently with one of our previous reports[20]).

Taken together, this illustrative case demonstrates the use of "fingerprint" representations. Notably, such fingerprint grid

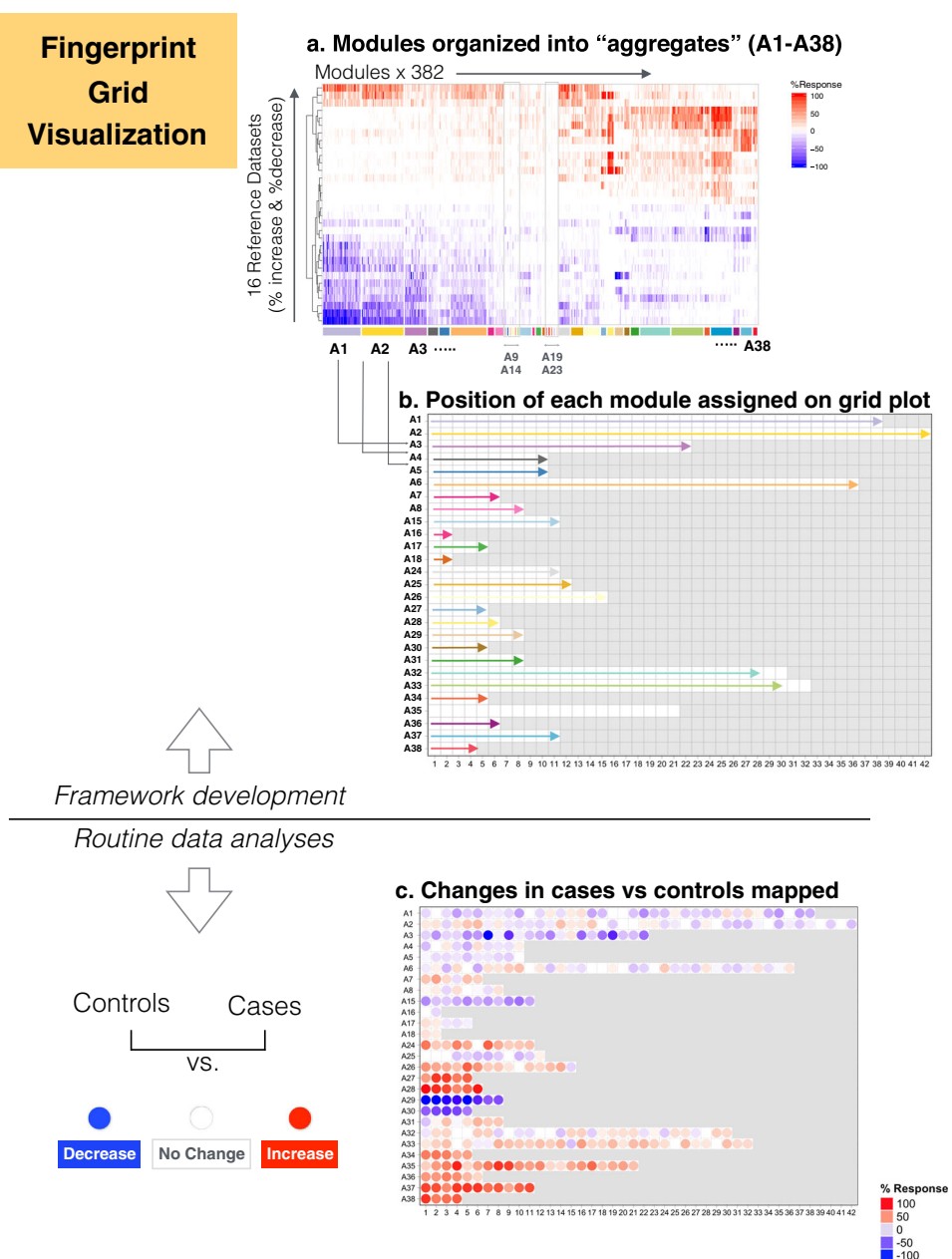

**Fig. 2 The development of the BloodGen3 module fingerprint grids. a** Rows on this heatmap correspond each to changes in transcript abundance for a given dataset and for a given direction (i.e. increase or decrease in transcript abundance). These values, the percentages of constitutive probes either increased or decreased within a module, are computed for the 16 datasets used as input for module construction (Table 1). Increases in transcript abundance compared to a healthy baseline are depicted in red and decreases are depicted in blue. Therefore, in total 32 rows are displayed on this heatmap. Columns correspond to modules comprising the BloodGen3 repertoire (N = 382). The colors shown on the bottom track are associated with module aggregate ID and only serve to illustrate the strategy that was employed for organization of modules on the fingerprint grid plot. **b** The modules were arranged onto the grid as follows: the master set of 382 modules was partitioned into 38 clusters (or aggregates) based on similarities among their module activity profiles across the sixteen reference datasets (A1–A38). A subset of 27 aggregates comprising two or more modules in turn occupied a line on the grid. The length of each line was adapted to accommodate the number of modules assigned to each cluster. The format of the grid was fixed for all analyses carried out using the BloodGen3 module repertoire. **c** When performing downstream analyses of blood transcriptome datasets using the BloodGen3Module R package changes in transcript abundance at the module level are mapped onto this grid and represented by colored spots of varying intensity.

representations are specific to a given module repertoire (i.e. BloodGen3 in this case). Fixing positions on the grid permits the overlay of functional annotations associated with each modular signature. This makes it possible for experienced users to "read" and interpret such fingerprints at a glance. It also permits the use of reference collections of BloodGen3 fingerprints for comparative interpretation. However, while this representation can be

used as a complement, it does not replace the more traditional heatmap representations.

**In-depth functional annotation of fixed transcriptional module repertoires.** Based on usage of earlier iterations (Gen1 & Gen2), the expectation is for this repertoire to be of use over a period

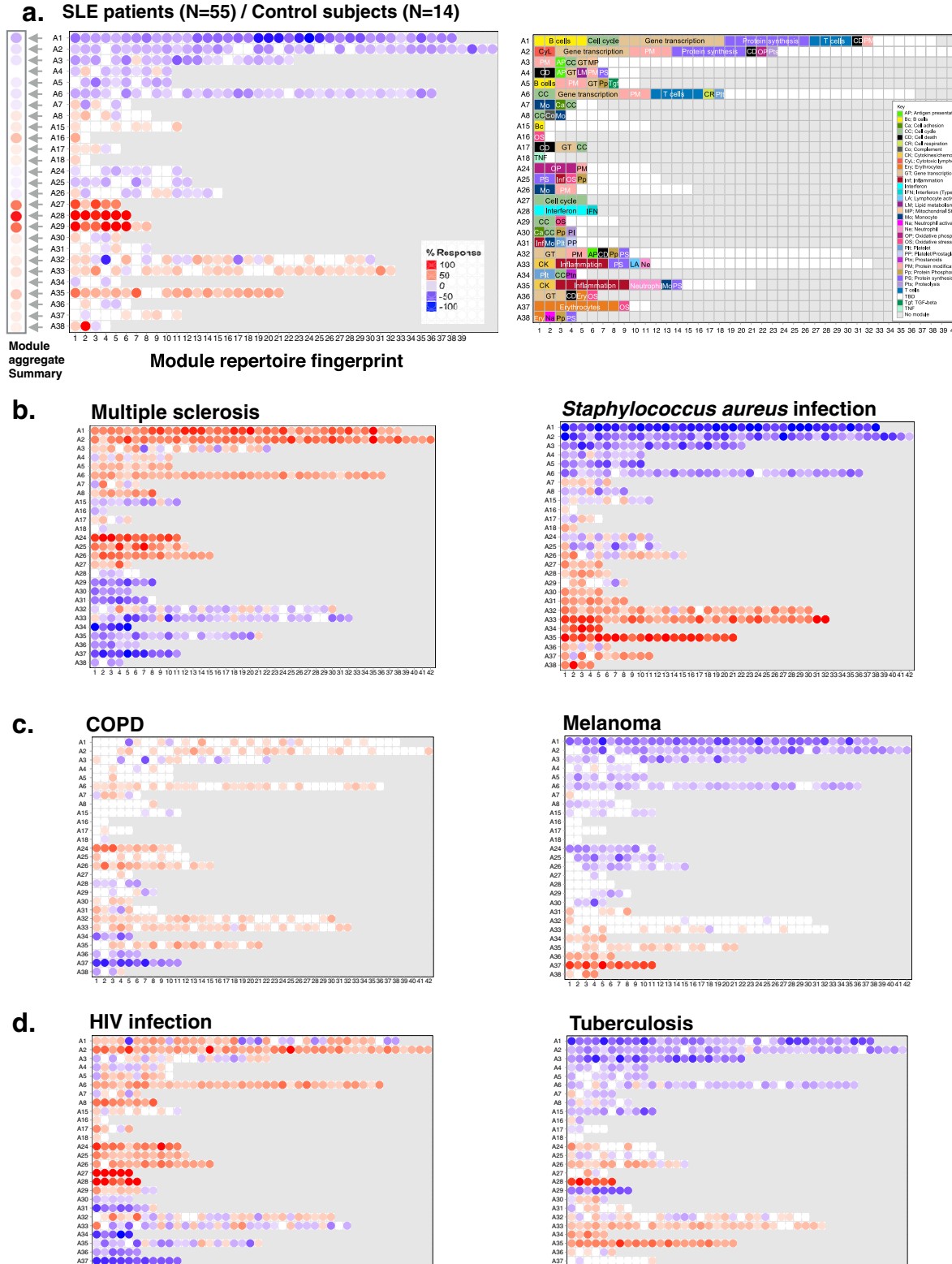

spanning several years. We therefore focused on comprehensive functional annotation of the BloodGen3 repertoire, and for this used two main approaches:[1] concurrent ontology, pathway or literature-term profiling analyses; and[2] determination for the constitutive genes for each module of expression patterns in select reference transcriptome datasets. We compiled the resulting information for BloodGen3 modules and made it accessible via

interactive circle packing plots. These interactive plots make it possible to zoom in and out, determine spatial relationships and interactively browse the very large compendium of analysis reports and heatmaps generated as part of our annotation efforts. (Fig. 4; links are listed in Supplementary Table 2 and are also accessible via the BloodGen3 web application: https://drinchai. shinyapps.io/BloodGen3Module/, and a demonstration video is

**Fig. 3 Fingerprint grid plots. a** Prototypical fingerprint grid plot. Changes in blood transcript abundance for patients with Systemic Lupus Erythematosus (SLE) compared to healthy controls are represented on a fingerprint grid plot for this illustrative case. The modules occupy a fixed position on the fingerprint grid plots (see Fig. 2). An increase in transcript abundance for a given module is represented by a red spot; a decrease in abundance is represented by a blue spot. Modules arranged on a given row belong to a module aggregate (here denoted as A1 to A38). Changes measured at the "aggregate level" are represented by spots to the left of the grid next to the denomination for the corresponding aggregate. The colors and intensities of the spots are based on the average across each given row of modules. A module annotation grid is provided where a color key indicates the functional associations attributed to some of the modules on the grid (top right). Positions on the annotation grid occupied by modules for which no consensus annotation was attributed are colored white. Positions on the grid for which no modules have been assigned are colored gray. **b–d** Fingerprint grid plots for additional reference datasets (COPD: chronic obstructive pulmonary disease, HIV: human immunodeficiency virus).

available at: https://youtu.be/db58FBUua-g). Below, we describe briefly how we conducted our annotations (see also Experimental Procedures):

*Step 1—Functional profiling:* We conducted gene ontology profiling for each of the 382 modules using DAVID[25], GOTERM, and GSAn[26]. GSAn interactive reports were uploaded to a custom web portal (https://ayllonbe.github.io/modulesV3/index.html). We also performed pathway enrichment analyses using the KEGG, BioCarta and the Ingenuity Pathway Analysis application, as well as literature-term enrichment with Literature Lab. Finally, we also used the RcisTarget tool in the R package to identify transcription factor binding motifs over-represented among the transcript constituents of each module[27]. We synthesized and compiled the results of the analyses (Supplementary Data 1) to identify convergences and attribute functional titles to the different modules. Functional titles could not be attributed in all cases due to a lack of convergence or poor enrichment in one or more of the analyses.

*Step 2—Expression patterns in reference transcriptome datasets:* We used transcriptome datasets as a reference to improve characterization and biological interpretation of the BloodGen3 module repertoire. Three different datasets were used. The first was contributed by Novershtern et al. and comprised the transcriptome profiles of 38 human hematopoietic cell populations[28]. The second was contributed by Speake et al. and comprised the RNA-seq profiles of six circulating leukocyte populations from patients with various immune-associated diseases[29]. The third was contributed by Monaco et al. and comprised RNA-seq profiles of 29 leukocyte populations isolated from healthy donors[30]. We generated heatmaps for each module to visualize the abundance patterns of the constituent transcripts for each dataset.

Overall, this resource serves two purposes. First, it provides access to information required to interpret the transcriptome fingerprints generated via module-level analyses. Second, it helps us to improve the accuracy with which functional titles and roles are attributed to the different modules and aggregates. Indeed, although the transcriptional module repertoire is fixed over time, we anticipate that the functional annotations will continue to evolve over its lifespan.

**Measuring inter-individual variability for the molecular stratification of patient cohorts.** The analysis and visualization steps presented so far focus on characterizing differences between groups of subjects (e.g. cases and controls). However, it is also important to characterize heterogeneity among groups of patients since inter-individual variability can serve as a basis for the definition of molecular phenotypes and patient stratification.

Within each module, and for each individual subject, we used fixed cut-offs to count the number of transcripts that increase, decrease or do not change in abundance compared to a baseline value (e.g. absolute fold change in expression and absolute difference in expression vs. average of control samples). The percentage of differentially expressed genes for each module is

then computed. These percentages are equivalent to values derived from group comparisons, except that they are derived for each individual sample.

The sepsis cohort included in the reference dataset collection was used to illustrate how this approach can be employed to assess inter-individual variability for a given pathology (Fig. 5). Changes in transcript abundance were found to be highly consistent across patients for some module aggregates. For instance, this was the case for aggregate A1 (broadly associated with lymphocytic cells/responses), with consistent decreases in transcript abundance observed across patients. Conversely, consistent increases were observed for modules comprising aggregate A35 (broadly associated with inflammatory neutrophil responses). In this case, differences were observed in the intensity of the response. However, other module aggregates, such as A37 (erythroid cells), A33 (functional association to be defined) and A28 (interferon response), showed more mixed responses.

A web application was developed as a resource to explore the inter-individual differences for a given disease, module aggregate or a combination of aggregates (BloodGen3Module App: https://drinchai.shinyapps.io/BloodGen3Module/; video: https://youtu.be/IXJDGeVH1bs). This application permits the generation of fingerprint grid plots and heatmaps representing module aggregate activity across the 16 reference datasets (Fig. 6).

This illustrative case focusing on sepsis shows that individual modular fingerprints can provide a means to achieve molecular stratification of patient cohorts. However, the biological and clinical relevance of such distinct molecular phenotypes would still remain to be determined in follow-on analyses.

**Profiling the abundance of A28 interferon-inducible genes at the aggregate level across reference patient cohorts.** Having explained the approach implemented for the construction and characterization of the fixed BloodGen3 transcriptional module repertoire, we now present the analysis and visualization strategies for both group- and individual-level comparisons using an illustrative case focusing on the changes in abundance for module aggregate A28 (interferon responses). We start from the highest possible perspective, examining changes in abundance for the A28 aggregate across reference disease cohorts.

A heatmap was derived showing patterns of abundance of a subset of 27 module aggregates comprising two or more modules across 16 health states (Fig. 7a). In the first order of separation, patients with acute HIV infection were grouped in one cluster, while the remaining 14 states were grouped into a second cluster. The main trend driving this dichotomy was an overall suppression of modules associated with inflammation and/or myeloid cell responses (A34–A38), accompanied by an increase in modules corresponding to aggregates associated, in part, with lymphocytic responses (A1–A8). The factors underlying these two distinct, "overarching" signatures are likely related to overall changes in myeloid versus lymphoid cell composition. Notably, diseases belonging to either group can exhibit marked interferon signatures (e.g. acute HIV infection in one cluster, and SLE or

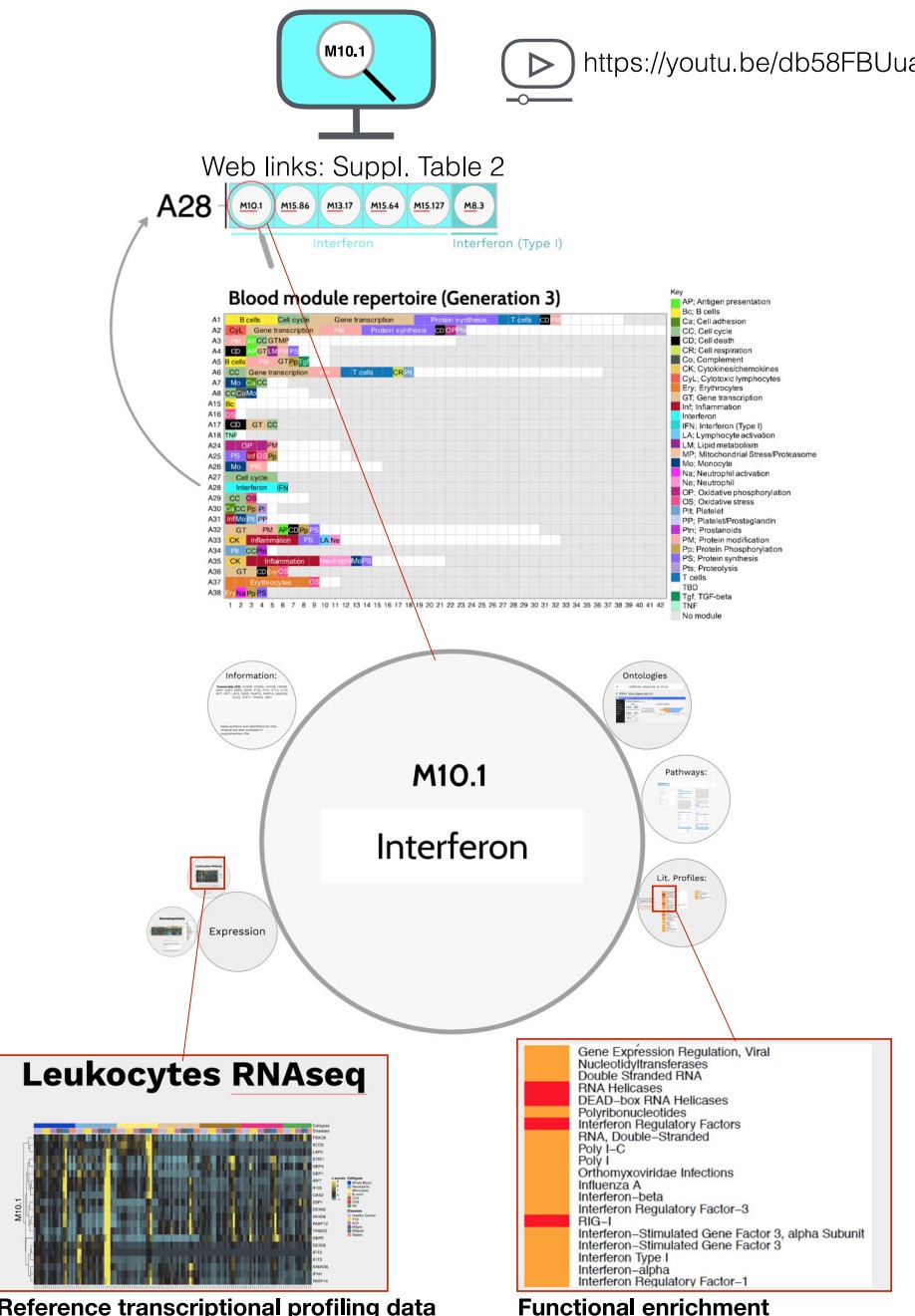

**Fig. 4 Functional annotation of the transcriptional module repertoire.** An interactive application is available to explore the 382 modules comprising the blood transcriptome repertoire. A gene list, along with the ontology, pathway, literature-term enrichment, and transcriptional profiling data for reference transcriptome datasets (circulating leukocyte populations, hematopoiesis) is provided for each module. Zoom in and zoom out functionalities for close-up examination of the text and figures embedded in the presentation are available. Web links providing access to modules within a given aggregate are listed in Supplementary Table 2. For a demonstration video, please visit: https://youtu.be/db58FBUua-g.

influenza infection in the other cluster). The circle plots shown in Fig. 7b–d provide a more granular illustration at the module and gene levels of the changes represented by spots on the heatmap. The gene composition for each of the six A28 modules is shown in circle plots in Fig. 7b. The circle plots shown in Fig. 7c highlight the genes changed in A28 modules when comparing patients with infection to their respective control groups (Fig. 7c). Finally, reference circle plots showing patterns of in vivo responses of A28 genes to administration of IFNα in patients with hepatitis C infection or of IFNβ in patients with MS [transcriptome profiling data were made publicly available by the authors[31,32]] are presented in Fig. 7d. The plots on this figure

show that changes observed at the aggregate level are not always distributed evenly across the six modules constituting aggregate A28, and in turn, of genes constituting each of the modules. The response to type I interferons was dominated by a disproportionate increase in abundance of transcripts constituting M8.3 and M10.1. In contrast, transcripts forming M15.86, which showed very little change in response to those treatments, were markedly increased during acute HIV and influenza infection (Fig. 7c). It is therefore possible that this gene set is more specifically induced by IFNγ. Interferon responses were weaker among RSV patients compared to these of patients with the two other viral infections. Similarly, in the context of bacterial

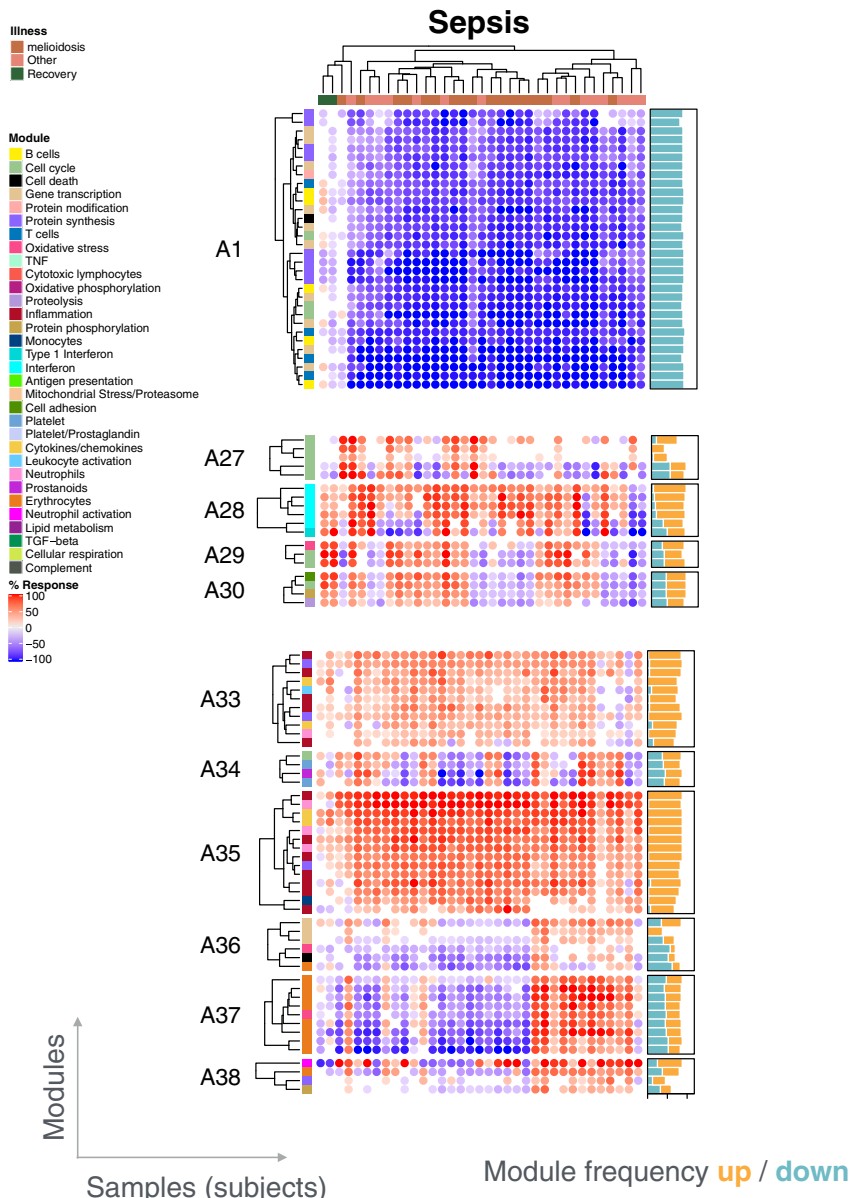

**Fig. 5 Individual-level module heatmap.** Changes in transcript abundance were determined at the individual level across all modules constituting the repertoire. These changes are represented on a heatmap, where an increase in abundance of a given module is represented in red, and a decrease in abundance is represented in blue. The subjects are organized as columns and the modules as rows. The order on the heat map was determined by hierarchical clustering.

infection, the interferon response was most marked in response to TB infection, which is consistent with previous reports[20]. A large proportion of adult patients comprising the sepsis cohorts were infected with *Burkholderia pseudomallei*, the intracellular bacteria responsible for melioidosis, which also tends to induce stronger interferon responses[22]. Changes in abundance of the transcripts constituting A28 in the context of autoimmune or inflammatory diseases are shown in Supplementary Fig. 2.

The visualization schemes employed here offer some new perspectives on the contribution of different modules and genes to the overall aggregate-level interferon responses. However, other variations in the selection of variables and granularity levels are possible and will be explored next.

**Profiling the abundance of A28 interferon-inducible genes at the module level across reference patient cohorts.** Plotting changes in the A28 genes across the same reference cohorts at the

module level, rather than at the aggregate level, provided a more granular perspective on the data. We also chose to compare literature keyword enrichment profiles for each of the A28 module at this level.

Functional enrichment analyses showed that all six modules in this aggregate were associated with the interferon response (see https://prezi.com/view/E34MhxE5uKoZLWZ3KXjG/). The heatmap (Fig. 8a) showing literature enrichment profiles highlighted keywords associated with viral pathogens ("hepatitis", "herpes" or "influenza"), as well as host-derived and pathogen-derived molecules ("RIG-I", "interferon", "interferon, "double-stranded RNA"). Among the six modules, four seemed to be "core" interferon modules (M15.86, M10.1, M8.3, M15.127), while the remaining two (M13.17, M15.64) were associated with the interferon literature to a lesser degree. These latter two modules were more strongly associated with the herpes simplex virus than the other four modules, while the four core modules were

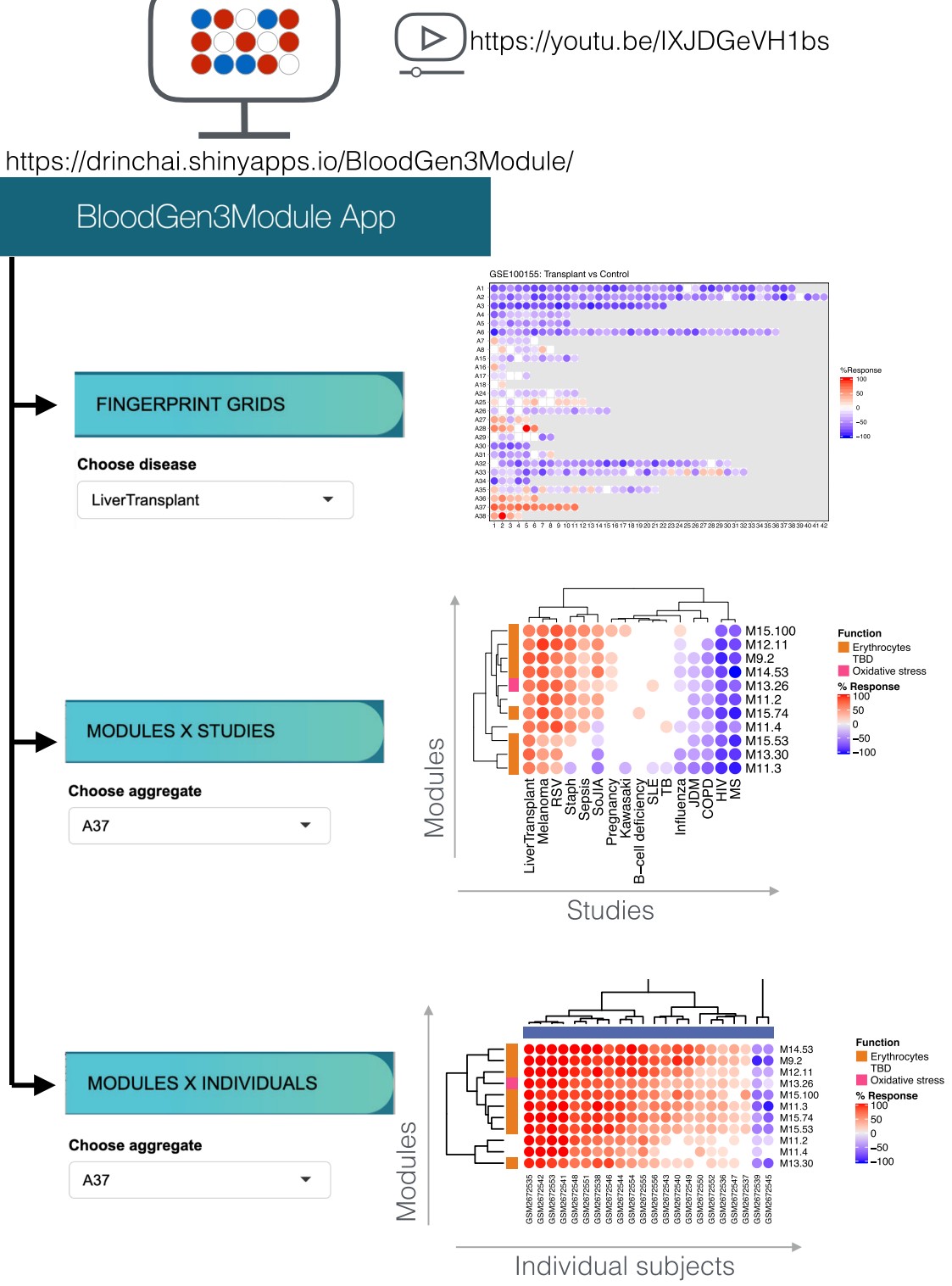

**Fig. 6 Web application to visualize multi-tiered module fingerprinting.** An application was developed to explore the changes in transcript abundance at the module level across the 16 reference datasets used to construct the repertoire. Three types of plot can be displayed and exported: (1) fingerprint grids; (2) module heatmaps displaying changes in abundance in modules comprising a given aggregate across the 16 reference datasets; and (3) module heatmaps displaying changes in abundance in modules comprising a given aggregate across individuals constituting a given dataset. To access the application, please visit: https://drinchai.shinyapps.io/BloodGen3Module/. For a demonstration video, please visit: https://youtu.be/IXJDGeVH1bs.

preferentially associated with hepatitis. The heatmaps available via the Prezi interface, which depicted the gene expression profiles for the A28 modules, provided additional perspectives. For example, the dataset from Speake et al. comprised samples of

patients with MS collected immediately before and 24 h after the administration of their first dose of IFNβ (Supplementary Fig. 3 [GEO ID GSE60424];[29]]. Despite the small number of subjects in this category, we observed a clear pattern of response to

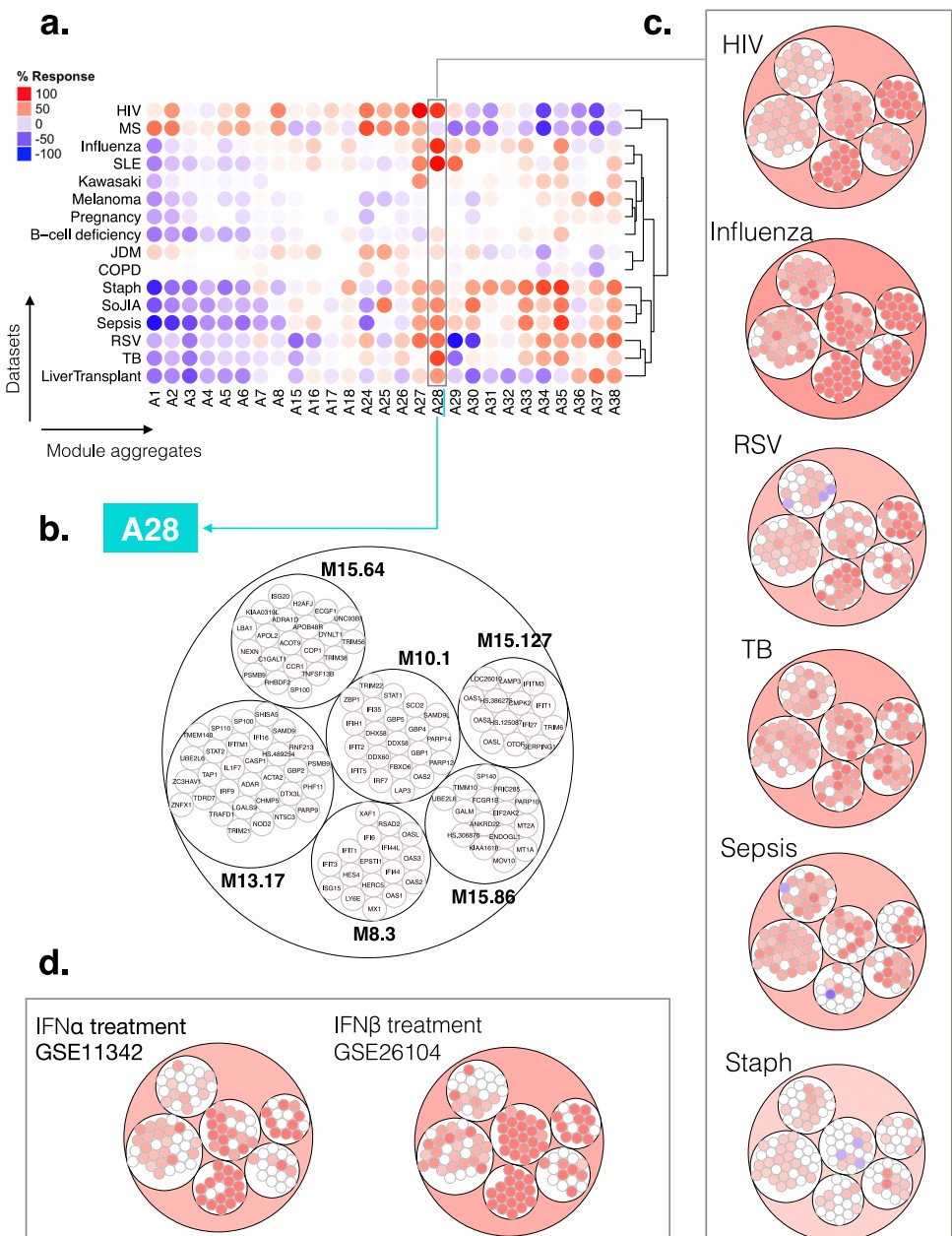

**Fig. 7 Module aggregate abundance patterns across the 16 disease or physiological states. a** Patterns of changes in transcript abundance at the aggregate and cohort levels. Each column on the heatmap corresponds to a "module aggregate", numbered A1 to A38. Modules A9–A14 and A19–A24 were excluded as they each comprised only one module. Each row on the heatmap corresponds to one of the 16 datasets used to construct the module repertoire. A red spot on the heatmap indicates an increase in abundance of transcripts comprising a given module cluster for a given disease or physiologic state. A blue spot indicates a decrease in abundance of transcripts. No color indicates no change. Disease or physiological states were arranged based on the level of similarity in the patterns of aggregate activity, determined via hierarchical clustering. **b** Representation of the modules and genes constituting aggregate A28. The circle plot represents the six modules constituting aggregate 28, and the transcripts constituting each of the modules. Some genes on the Illumina BeadArrays can map to multiple probes, which explains the few instances where the same gene can be found in different modules. **c** Patterns of changes in transcript abundance at the module level and gene level for aggregate A28. The circle plots illustrate the changes at the gene level for this aggregate for 6/16 datasets. The position of the genes on each of these plots is the same as shown in panel B. Genes for which transcript abundance is changed are shown in red (increase) or in blue (decrease). **d** Patterns of changes in transcript abundance at the module and gene levels for aggregate A28 in subjects treated with IFN-α or IFN-β. The circle plots show changes in abundance of A28 transcripts in patients with hepatitis C infection treated with IFN-α [GSE11342[31]] or patients with MS treated with IFN-β [GSE26104[32]] (HIV: human immunodeficiency virus, RSV: respiratory syncytial virus, TB: Tuberculosis, Staph: *Staphylococcus aureus* infection, SLE: systemic lupus erythematosus, MS: multiple sclerosis, JDM: juvenile dermatomyositis, COPD: chronic obstructive pulmonary disease, SoJIA: systemic onset juvenile idiopathic arthritis, IFNα: interferon alpha, IFNβ: interferon beta).

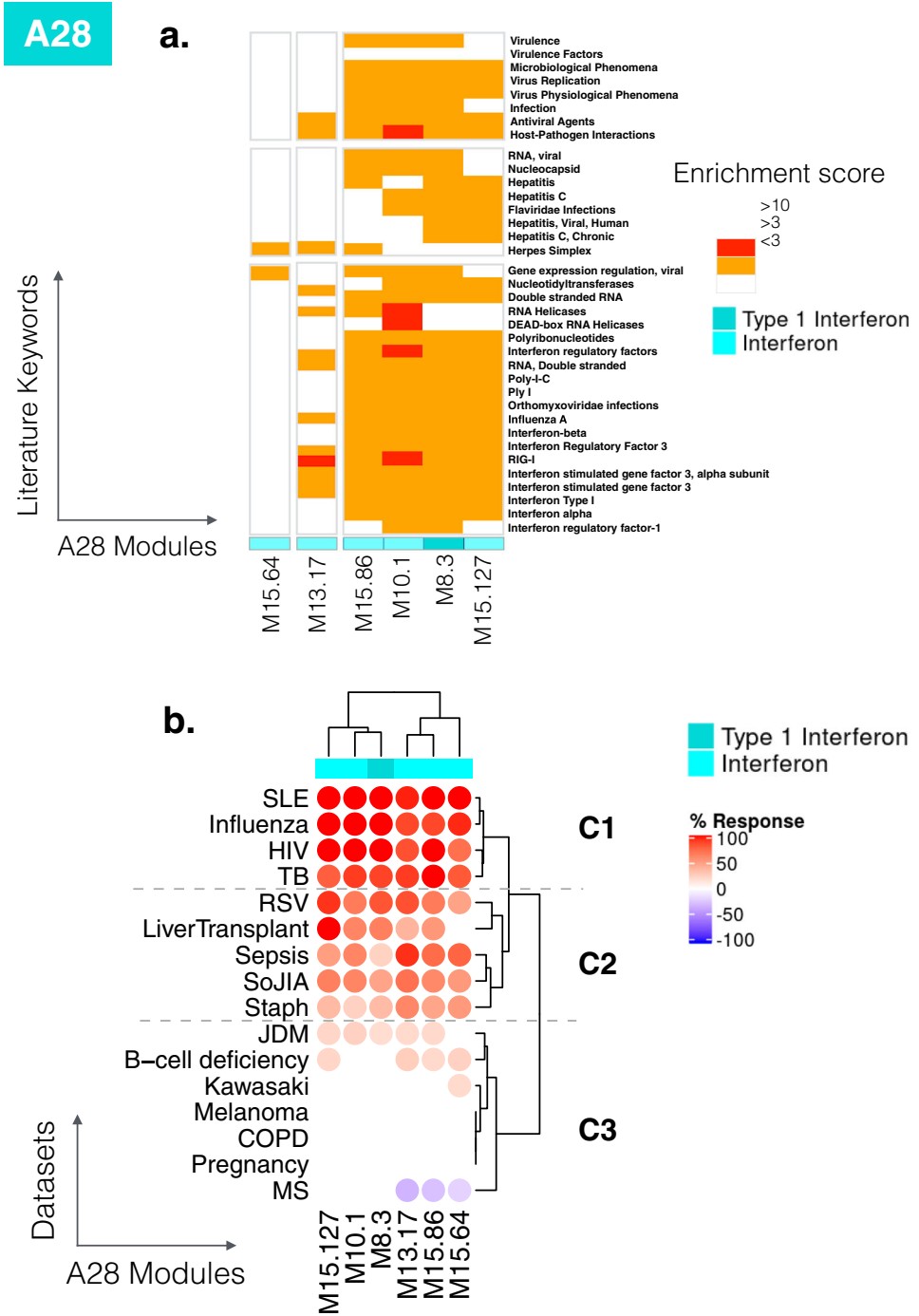

**Fig. 8 Literature profiles and patterns of changes in abundance across reference datasets for the modules comprising aggregate A28. a** Functional annotation by literature profiling. A. Portion of a heatmap comprising 382 modules organized as columns, and literature terms organized as rows. The six modules shown are associated with the consensus annotation "Interferon" or "Type 1 Interferon". The clusters of keywords associated with those modules are consistent with this annotation and provide added granularity to the module repertoire for functional profiling and interpretation. **b** Changes in abundance across 16 reference datasets. The heatmap represents the changes in abundance of transcripts constituting the six modules comprising aggregate A28 (columns). The modules are functionally associated with interferon responses. The 16 reference datasets are arranged as rows corresponding to different health states. The columns and rows are arranged by hierarchical clustering. The heatmaps can be accessed and exported for all 16 datasets and 38 module aggregates using the web application: https://drinchai.shinyapps.io/BloodGen3Module/ (under the "MODULES X STUDIES" tab).

interferon in vivo across all six modules. This observation confirmed the functional associations identified in the enrichment analyses.

Next, we examined the degree of changes for the six interferon modules across the 16 input datasets (Fig. 8b). The first cluster

showing the highest induction levels comprised SLE, influenza infection, HIV infection and active *M. tuberculosis* infection (labeled C1 on the figure). Interferon has antiviral properties; therefore, it was no surprise to see viral infections included in this set. Blood transcriptome profiling studies conducted nearly 20

years ago revealed interferon responses in the pathogenesis of SLE[33,34]. More recent profiling also revealed the prominence of this signature in patients with tuberculosis, which contrasts with findings made in other bacterial infections[20]. The second cluster (labeled C2) comprised diseases with an "intermediate" level of interferon responses, including RSV infection, sepsis caused by *Staphylococcus aureus* in pediatric patients and a range of bacterial pathogens in adults, SoJIA and liver transplant recipients receiving maintenance immunosuppressive therapy. The illness conferred by RSV infection has a clinical presentation very similar to that of influenza in pediatric patients. However, studies suggest that interferon responses may be partially inhibited by RSV[35,36].

The third and final cluster was formed by pathologic and physiological states where an increase in abundance of interferon-inducible genes was either modest or non-existent. This cluster included patients with juvenile dermatomyositis and B-cell deficiency (low levels), Kawasaki disease, melanoma, COPD, pregnancy (no increase), and individuals with MS (apparent decrease). The latter observation is interesting since IFNβ administration is one form of treatment for MS, which should compensate for the defect observed here in treatment-naïve individuals.

Taken together, this closer examination of annotations for A28 modules and patterns of changes in abundance in different reference datasets provided a clearer picture of its biological significance.

**Profiling the abundance of A28 interferon-inducible genes at the module level across individual subjects.** Analysis workflows were developed to determine changes in transcript abundance at the level of individual subjects. This approach, which offers an even more granular perspective, was next applied for molecular stratification of patient cohorts using the six A28 interferon modules.

We showed, for instance, that in cohorts where no changes were detectable via comparisons at the overall group level, the signature was in fact presented by a minority of patients. Indeed, this scenario applied for the cohort of melanoma patients with three of 22 subjects showing some degree of increase in abundance for the six A28 modules, while the majority showed small changes or a decrease in abundance. At least four of the patients showed a marked decrease in abundance (Fig. 9a). This observation is of potential biological and clinical significance as interferon activity in patients with melanoma has been associated with disease outcomes previously[37,38]. Pathologies with an intermediate A28 signature might induce responses in a higher proportion of subjects. This was the case for the JDM patient cohort comprising a cluster of 11 of 40 patients presenting with modular interferon signatures (Fig. 9a). Of the 793 articles in PubMed mentioning "juvenile dermatomyositis" in the title, eight also mentioned "interferon", indicating that while not widely acknowledged, a role for interferon in this disease has nonetheless been described[39]. In diseases for which the role of interferon is well-described, such as influenza infection or SLE, increases in A28 modules were widespread, although the interferon signature was not detected in a few subjects in each cohort (Fig. 9a). Notably in the case of SLE, the proportion of interferon-negative subjects tended to be higher in adult patient cohorts in comparison to pediatric patient cohorts such as that is being used for illustrative purposes here. We next present stratification of an adult SLE cohort based on patterns of abundance of A28 modules.

Using our second-generation repertoire, we have previously shown that the interferon signature characterizing SLE comprises distinct "sub-signatures" at the module level[24]. We also observed a sequential increase in a set of three second generation interferon modules (M1.2, M3.4, and M5.12). M1.2 showed a higher degree of sensitivity, followed by M3.4 and then M5.12. Based on this finding, we stratified SLE patients based on whether one, two, or all three of those interferon modules were activated. By combining functional profiling with a reference dataset, we concluded that the modules responded differently to each interferon type: IFN-α induced an increase in the abundance of genes belonging to M1.2; IFN-β induced an increase in the abundance of genes belonging to M1.2 and M3.4; and interferon-γ induced an increase in the abundance of genes belonging to M5.12. Next, we sought to determine the equivalence between these three, second-generation interferon modules and the six new, third-generation interferon modules regrouped in aggregate A28. Based on gene composition, M8.3 and M15.127 mapped to M1.2 (inducible by both IFN-α and -β), M10.1 and M15.86 mapped to M3.4 (inducible by IFN-β), and M13.17 and M15.64 mapped to M5.12 (inducible by IFN-γ). Notably, the latter two modules did indeed segregate from the other four based on the literature enrichment profiling heatmap shown in Fig. 8a. We also used the six interferon modules to re-classify the adult SLE dataset profiled in our earlier study [GEO ID GSE49454[24]]. The resulting clustering and stratification mirrored our earlier findings made using the three interferon modules (Supplementary Fig. 4). Overall, these observations confirmed that interferon "sub-signatures" can be employed for patient stratification. This may be relevant in the tailoring of biologics targeting interferon in development for the treatment of SLE. However, the potential of utilizing the six modules from the new repertoire to improve on the three from the second-generation repertoire employed previously remains to be determined.

Next, we aimed to illustrate how further insights can be obtained by examining changes in A28 modules abundance at the individual level concurrently with those of other module aggregates. As an example, we present the changes for both A28 (interferon) and A35 (inflammation) modules (Fig. 9b). By combining the two modular signatures, we assessed their relative contributions in a given cohort and identified distinct phenotypes: i.e. interferon or inflammation "positive", "double-positive", or "double-negative". We found the most contrasting patterns of changes in transcript abundance in the sepsis and melanoma cohorts. Here, the abundance of interferon (A28) and inflammation (A35) modules almost uniformly increased in patients with sepsis. In contrast, we observed increases in these modules in only a minority of melanoma patients, with increases in inflammation modules being more widespread than those in interferon modules (approximately 50% compared to approximately 10% of subjects, respectively). Parallel findings were obtained in pregnancy, which represents another immunosuppressive state, although increases in abundance of a subset of A35 modules appeared to be more prevalent in this group than in melanoma patients. The pattern of changes in influenza infection was more similar to that in sepsis, but with higher levels of interferon induction and a lower level of inflammation.

Overall, the illustrative case presented here demonstrates the preliminary stepwise dissection of a given module aggregate and investigation of the underlying biological relevance. This process could be repeated for other module aggregates and while this is beyond the scope of the present study, it is a process that will support the scalable annotation infrastructure that has been developed here. Indeed, we expect the range of reference datasets and functional profiling approaches available for interpretation to continue to expand and the processes described here should also help determine to what extent subdivision of signatures in distinct modules is warranted.

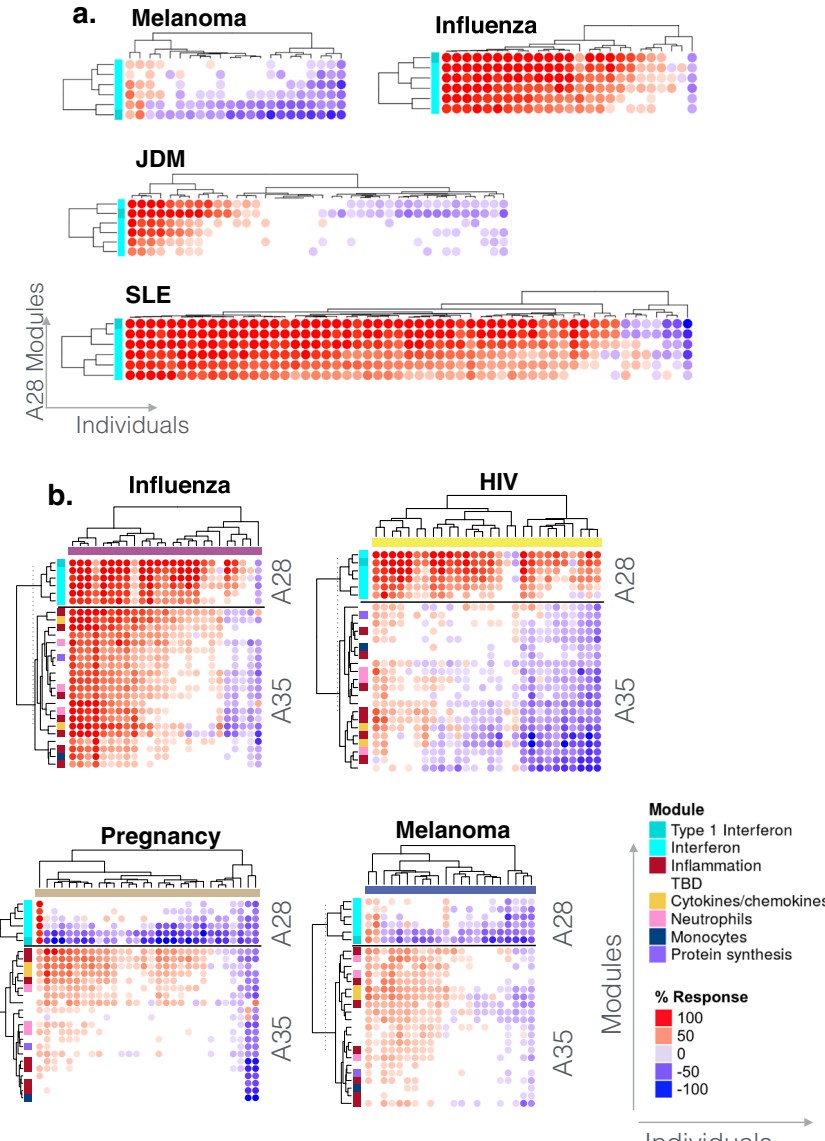

**Fig. 9 Abundance patterns across individuals. a** Changes in abundance for A28 modules. The heatmaps display the changes in abundance for the same six modules (rows) across individuals (columns) in four reference cohorts. The rows and columns on the heatmap are arranged based on similarities in abundance patterns. **b** Changes in abundance for A28 and A35 modules. The heatmaps display the changes in abundance of six modules constituting aggregate A28 and 21 modules constituting aggregate A35 (rows) across individuals (columns) in four reference datasets. Functional annotations associated with different modules are indicated by a color code and corresponding legend. The heatmaps can be accessed and exported for all 16 datasets and 38 module aggregates using the web application: https://drinchai.shinyapps.io/BloodGen3Module/ (under the "MODULES X INDIVIDUALS" tab).

**Development and availability of ancillary resources.** BloodGen3 modules can be reused as a framework for the analysis of blood transcriptome datasets which were not originally included for the constitution of the repertoire. We have developed custom bioinformatic resources to support downstream analyses visualization and interpretation carried out specifically using the BloodGen3 repertoire. These resources have been employed in several published use cases, all of which involving the analysis of multiple public datasets which were not employed for the construction of the BloodGen3 repertoire: (1) In one instance, BloodGen3 modules served as a framework for the development of targeted blood transcript panels (in the context of COVID-19[40]); (2) Another illustrative case involved the use of the repertoire and associated interpretation resources described here to identify and characterize a systemic signature of psoriasis disease, and screen for constitutive genes targeted by existing drugs[41]. (3) In the final published illustrative case, the BloodGen3 module repertoire was employed as a framework for an integrative meta-analysis encompassing six independent RSV datasets[42].

An R package that we have developed to support BloodGen3-based data analysis and visualization (BloodGen3Module), was employed to carry out the analyses presented in these illustrative use cases and the current work. This tool has been described in detail in a separate publication[43] and scripts are openly available via GitHub (https://rdrr.io/github/Drinchai/BloodGen3Module/). Briefly, this resource includes functions that can be used to perform differential expression analyses and visualize datasets generated by the end users as fingerprint grid plots or heatmaps.

Other project-specific bioinformatics resources that were developed include a web application that provides users with the ability to generate custom fingerprints plots (grids or

heatmaps), presently for the 16 reference cohorts employed for the construction of the repertoire (BloodGen3 app: https://drinchai.shinyapps.io/BloodGen3Module/). Different configurations can be selected for the heatmap, for instance to display fingerprints for modules within a given aggregate across the 16 reference datasets, or across all the individuals within a given dataset. For the latter application, it is also possible to select and combine multiple aggregates on the same heatmap. The resulting plots and corresponding tables can be exported and employed to prepare figures or generate plots using other tools. The Blood-Gen3 web application also integrates links to Prezi circle packing plots generated for each of the aggregates, thus presenting users with more unified access to supporting resources. Two additional project-specific web applications with similar functionalities have been developed. One was dedicated to the published COVID-19 use case mentioned above (https://drinchai.shinyapps.io/COVID_19_project/[40]) and the second to the respiratory syncytial virus infections use case (https://drinchai.shinyapps.io/RSV_Meta_Module_analysis/[42]).

Altogether these resources complement the extensive annotation framework presented earlier (interactive Prezi circle packing plots), to provide potential users means to leverage the Blood-Gen3 repertoire for the analysis of their own blood transcriptome datasets.

## Discussion

The BloodGen3 module repertoire is meant to be employed as a fixed and reusable framework for the analysis of blood transcriptome data. It has been constructed through co-expression analyses carried out across a wide range of diseases and physiological states. It constitutes the third iteration released by our group since 2008. Another blood transcriptome repertoire has been developed and made available by our collaborators from Emory University[23]. Several differences in the approach employed for the construction of this latter 334-module repertoire should be noted. First, the development of the repertoire reported by Li et al. relied on public transcriptome datasets ($N >$ 500), and second, selection of constitutive genes for the modules relied partly on co-expression and partly on functional convergences (ontology category, cell type-specific expression, interactome or bibliome). More recently, Zhou and Altman described the development of another fixed transcriptional module repertoire based on the assembly of a collection of 2753 public datasets that were deposited in the NCBI GEO and encompassed 97,049 unique transcriptome profiles[44]. Independent component analysis was then used to resolve co-expression relationships, which led to the identification of a set of 139 transcriptional modules. Subsequently, the biological relevance of these sets and advantages of their use for improving the robustness of analyses, especially for smaller datasets, were presented. Overall, this provided additional arguments in favor of the reuse of fixed transcriptional module repertoires, a practice that, to date, has not been widespread. However, Zhou and Altman chose datasets generated from a wide range of sample types as input data and the resulting framework may not capture some of the specificities that exist within more narrowly defined biological systems (for example a tissue or cell type such as the blood transcriptome in the case of the BloodGen3 repertoire). It is also worth noting here that large collections of gene sets are commonly used for the interpretation of transcriptome profiling data [e.g., gene set enrichment analyses (GSEA):[45,46]]. Such reference collections are very large, typically numbering tens of thousands of signatures. While these datasets have proven useful as a reference for functional interpretation, they have not been developed with a focus on a specific biological system or application, or to perform reductive analyses.

Transcriptional module repertoires are also commonly constructed for "single use", using popular approaches such as whole-genome co-expression network analysis (WGCNA)[47]. Such repertoires are based on and used for the analysis of a given dataset and are not designed to be reused, as is the case for the "fixed" module repertoires described above. Notably, the continual reuse of fixed module repertoires over time periods usually spanning several years means that greater effort and resources can be dedicated to the development of tailored repertoire-specific analytic resources, as exemplified here in the case of BloodGen3. We previously attempted to develop such a support infrastructure for our second-generation of modules, albeit on a smaller scale. However, we learned that maintaining such resources over long periods of time proved a challenge as these now have gone offline through a combination of hardware failures, hacking or discontinued institutional support. Here we attempted to address these issues through adoption of zero cost infrastructure (e.g. depositing of R scripts in GitHub, deploying R Shiny apps, uploading videos to YouTube, or employing the Prezi platform that makes presentations freely accessible as long as public access is maintained).

These resources include a R package, "BloodGen3module", that can be used to generate fingerprint grid plots and heatmaps[43]. These custom visualizations have been developed to support the interpretation of blood transcriptome data expressed at the module level. Notably, the two-tiered grouping, first at the module level and at the module aggregate level, is new for this third-generation module repertoire. This in turn led to adopting a different strategy for fingerprint grid plots visualization. Indeed, for earlier repertoires (Gen1 & Gen2), the order in which modules were selected determined their order on a given row of the grid (first to last, from left to right). For the BloodGen3 repertoire, modules within a given aggregate (=row) are grouped first according to functional annotations and then among the different annotations, from left to right according to alphabetical order. The modules without annotations are added last and ordered in ascending order according to their module ID (i.e. based on the selection process).

It may also be worth noting that changes in relative cell composition will in part be driving fluctuations in transcript abundance in whole blood. As such, co-expression analyses and the construction of module repertoires will permit the identification of gene sets associated with specific leukocyte populations. Indeed, functional annotations and reference leukocyte profiling datasets unequivocally associated a number of BloodGen3 modules with cell types (e.g. B-cells, T-Cells, cytotoxic cells, neutrophils or erythroid precursors). A selection of the 382 modules comprising this repertoire could therefore be employed specifically for bulk blood transcriptome data deconvolution and estimation of changes in abundance of those cell populations. This is along the lines with what tools such ABIS[30], or CIBERSORT are offering. A notable difference being that relying on module repertoires does not require making a priori choices regarding which leukocyte populations should be deconvoluted from bulk transcriptome profiling data. One of our recently published illustrative cases highlights this aspect in that it describes a dominant circulating erythroid cell signature (A36, A37, A38) with putative immunosuppressive function that was found to be associated disease severity in patients with respiratory syncytial virus infection[42], but that is not otherwise included in the panels of cell populations commonly used for deconvolution of blood transcriptome profiling data.

Finally, limitations inherent to the BloodGen3 repertoire and associated resources shared here are also worth noting: First, the

BloodGen3 repertoire is specifically designed to analyze and interpret human blood transcriptome profiling data. Analyses of other tissues or in the context of other species would require the development and use of separate frameworks. Indeed, we have also been involved in the development of fixed module repertoires for various mouse tissues[48] and in vitro culture systems, including human whole blood[49] and human dendritic cells[50].

Second, it should be also noted that the BloodGen3 repertoire could be altered by the addition or removal of datasets used for its construction. However, we do not expect many changes at the least granular "module aggregate level". Indeed, the dominant functional themes found in BloodGen3 aggregates (plasma cells, interferon, inflammation, cytotoxic/T-cell responses, erythrocytes etc.) were already present in both Gen1 and Gen2. More variations would be expected at the more granular module level. We have previously found these inter-module differences to be biologically meaningful[24], and conserved from Gen2 to Gen3 in the case of interferon.

Third, we generated our transcriptome datasets for module construction using Illumina BeadArrays, a technology that predates RNA sequencing. Our choice to use this technology was based on availability at our institute for the construction of the BloodGen3 repertoire of a collection of well-characterized blood transcriptome datasets generated at the same facility and obtained across different projects using harmonized protocols. An alternative approach would be to compile publicly available blood transcriptome for which appropriately matched control data are available. The coverage afforded by RNA-seq data would likely result in somewhat larger modular gene sets; however, we consider it unlikely that entire co-expressed gene sets would be missed by microarrays to the extent that it would lead to the production of a repertoire that is fundamentally different to that produced using RNA-seq data. It is also worth noting that the BloodGen3 repertoire is currently routinely used in our laboratory to perform analyses using RNA-seq data (some illustrative cases are already available[40,41]). In addition, the "Gen2" repertoire, also based on whole blood and Illumina BeadArrays[17], has been used to analyze profiles generated from peripheral blood mononuclear cells using a targeted 700-transcript Nanostring panel[51]. It is worth drawing the attention of investigators who wish to employ the framework for the analysis of RNA-seq data that the BloodGen3 repertoire was constructed at the probe level (those present on the Illumina HT12 v3 Beadarrays). This was to account for the possibility that changes in abundance for transcript variants measured by distinct probes may differ. And indeed, in some cases different probes mapping to the same genes were distributed across several modules (often within the same aggregate). Thus, for those analyzing data generated at the gene level via RNA-seq the choice would be to assign the same value to duplicate genes found in different modules. Another possibility would be to disregard genes with duplicates altogether, which in our hands shows to have very little impact on the results since they concern only a small minority of genes (388 out of 14,168 probes). Both options can be implemented using functions provided within the BloodGen3Module R package[43].

Fourth, our module repertoire is not meant to constitute a ground truth. Indeed, some aggregates are clearly not homogeneous in terms of functionality, and heterogeneity also exists within modules. The attribution of functional annotation titles to modules also confers some degree of subjectivity. However, the repertoire does structure the data so that insights about the biological significance of such a modular signature can be determined. As a result, while the framework presented here will remain fixed for at least the next few years, it is likely that the functional annotation map will continue to evolve for the foreseeable future.

In conclusion, the approach to the development of module repertoires and associated interpretation resources described here should be generalizable to other biological systems (different sample types, species or data types). For instance, possible applications would be the development of reusable repertoires based on transcriptome profiles generated from other tissues or cell types. In addition, more narrowly defined blood transcriptome repertoires may also be derived (e.g., based on blood transcriptomes for a given disease or set of diseases). Notably, one of the main bottlenecks in the development of such resources would likely be the development of the companion analysis and interpretation frameworks, rather than the construction of the repertoire per se. It should be possible to partly automate and streamline the annotation process, and the construction of interactive circle packing plots for instance, although a manual component will likely remain when drawing functional inferences (i.e. "connecting the dots"). Nevertheless, as is expected of the BloodGen3 module repertoire, once initially established, such interpretation frameworks may also be developed over time as more analyses are being carried using such repertoires.

## Methods

**Study subjects.** Gene expression datasets from 985 de-identified subjects from distinct cohorts were used for this research. Written informed consent was obtained from all participants. Studies were approved by Institutional Review Boards of the Baylor College of Medicine (COPD dataset: H-18029), the University of Texas Southwestern Medical Center and Baylor Health Care System (Influenza, RSV, S. aureus and Kawasaki disease datasets: UTSW #0802-447/BIIR #002-141), Saint Jude's Research Hospital (B-cell deficiency), the Baylor Health Care System (Liver transplant: 002-197, Pregnancy: 009-257, Multiple sclerosis: 009-240, Melanoma: 006-025 & 097-027), Khon Kaen University (Sepsis), the University of Texas Southwestern Medical Center (SoJIA, Dermatomyositis, SLE), Duke University and the Baylor Health Care System (HIV: Duke 8485-06-4R0/Baylor 006-177), St. Mary's Hospital London, UK and University of Cape Town, Cape Town, Republic of South Africa (Tuberculosis: St Mary's REC 06/Q0403/128, University of Cape Town REC 012/2007). The gene expression datasets selected to cover major classes of immune states (Table 1) were required to have at least 25 samples in total, and at least 20% of the total samples were required to be controls matched for gender, age and ethnicity.

General descriptions of the study cohorts are as follows: S. aureus cohort: Children with community-acquired S. aureus infection were enrolled. The clinical syndromes of these patients included skin and soft tissue infection, bacteremia, osteomyelitis, suppurative arthritis, pyomyositis, pneumonia, and disseminated disease. Patients diagnosed with toxic shock syndrome, polymicrobial infections, or treated with corticosteroids in the preceding four weeks were excluded; Adult sepsis cohort: Diagnosis of sepsis was based on accepted international guidelines and defined as presentation with two or more of the following criteria for the systemic inflammatory response syndrome: fever (temperature > 38 °C or <36 °C), tachycardia (heart rate >90 beats/minute), leukocytosis or leukocytopenia (white blood cell count $\geq 12 \times 10^9/l$ or $\leq 4 \times 10^9/l$). Blood was collected within 24 h following the diagnosis of sepsis. Samples were selected for microarray analysis from subjects with a diagnosis of bacteremic sepsis retrospectively confirmed by the isolation of a causative organism on blood culture; factors accounted for in the selection of subjects in the control group included gender, age and type 2 diabetes diagnosis, the latter being a risk factor for septicemic melioidosis. TB cohort: Patients were prospectively recruited and sampled, before any anti-mycobacterial treatment was initiated. Active TB disease was confirmed by laboratory isolation of M. tuberculosis on mycobacterial culture of a respiratory specimen (either sputum or bronchoalveolar lavage fluid); Influenza cohort: Children with confirmed influenza infection were recruited. Those with documented bacterial co-infections or chronic conditions and systemic steroid treatment within 2 weeks before enrollment were excluded; RSV cohort: Children with confirmed RSV infection were recruited. Children with documented bacterial co-infections, congenital heart disease, chronic lung disease, immunodeficiency, prematurity (<36 wk), systemic steroid treatment within 2 weeks before presentation or additional chronic comorbidities were excluded; HIV cohort: Blood samples were obtained from adult patients diagnosed with HIV infection. At enrollment patients were verified as acute Fiebig stages 4–6 (plasma RNA +, third-generation EIA+, Western blot indeterminant or + ); SLE cohort: Blood samples were obtained from pediatric patients diagnosed with systemic lupus erythematosus and healthy controls matched for demographic characteristics; MS cohort: Subjects enrolled in the MS cohort had an established diagnosis of relapsing-remitting MS, separately confirmed by an experienced MS neurologist (JTP), exhibited no other health conditions, and had received no treatment(s) for MS, including corticosteroids, for at least 3 months prior to blood collection; Juvenile dermatomyositis cohort: Blood samples were obtained from pediatric patients diagnosed with juvenile

dermatomyositis and healthy controls matched for demographic characteristics. Kawasaki disease cohort: Subjects <18 years of age who met the definition of Kawasaki disease based on the American Heart Association (AHA) criteria[52] were enrolled alongside age- and gender-matched healthy controls; Systemic onset juvenile arthritis cohort: Blood samples were obtained from SoJIA patients displaying systemic symptoms only (fever, rash, and/or pericarditis) or displaying systemic symptoms accompanied by arthritis; COPD cohort: Enrollment criteria included age over 40 years, no history of concurrent lung cancer, chest surgery, or chronic lung diseases other than COPD (e.g., sarcoidosis, fibrosis, etc.). Participants had no history of allergies or asthma and at the time of initial recruitment had not received oral or systemic corticosteroids during the previous 6 weeks; volunteers were enrolled from three clinics within the Texas Medical Center in Houston (TX, USA); B-cell deficiency cohort: This cohort comprised adults with diagnosis of XLA as documented by markedly reduced numbers of peripheral blood B-cells; Pregnancy cohort: Pregnant women were recruited at the Baylor Institute for Immunology Research (Dallas, TX, USA) for a study of immunological signatures of pregnancy; Melanoma cohort: Enrollment criteria included age 21–75 years, stage M1a, M1b, M1c biopsy proven metastatic melanoma patients with measurable metastatic lesions by physical examination or scans, acceptable CBC and blood chemistry results, adequate hepatic and renal function, and no active CNS metastatic disease; Liver transplant cohort: Enrollment criteria included age 17–65 years, having received a liver transplant under maintenance immunosuppression therapy. Subjects in this cohort had not received an acute or chronic rejection diagnosis at the time of sampling.

**RNA extraction and processing**. Whole blood for all sample sets was collected into Tempus Blood RNA Tubes (Thermo Fisher Scientific, Waltham, MA, United States). Total RNA was isolated from whole blood lysate using a MagMAX for Stabilized Blood Tubes RNA Isolation Kit for Tempus Blood RNA Tubes (Thermo Fisher Scientific). RNA quality and quantity were assessed using an Agilent 2100 Bioanalyzer (Agilent Technologies, Santa Clara, CA, United States) and a Nano-Drop 1000 (NanoDrop Products, Thermo Fisher Scientific). Samples with RNA integrity number values >6 were retained for further processing.

**Microarray analysis and data preprocessing**. Gene expression profiles from whole blood samples generated using Illumina HumanHT-12 v3.0 expression BeadChips were obtained from 16 groups of patients and controls selected as above. Thus, 16 datasets were used as the input data (Table 1). The expression data for each dataset were preprocessed and independently clustered. First, the probes were discarded if they were not detected (detection $P < 0.01$) in at least 10 samples or in at least 10% of the samples, whichever was greater. Then, the sample data for each dataset were normalized using the BeadStudio average normalization algorithm. Once normalized, the signal was transformed such that all signals <10 were set to 10. Then, the fold change was calculated relative to the median signal for that probe across all samples. If the difference between a signal and the probe's median signal was <30, or the calculated absolute magnitude of the fold change was <1.2, the fold change was set to 1 to reduce the noise from low-level responses. At this stage, the probes were filtered again. Probes were only retained if they had a calculated absolute fold change >1 in at least 10 samples or in at least 10% of the samples, whichever was greater. Finally, the data were transformed to the $\log_2$ of the calculated fold changes.

**Module construction algorithm**. Sets of coordinately regulated genes, or transcriptional modules, were extracted from the patient's whole blood microarray datasets. Full details and an example of the code are included in the supplemental methods (Supplementary Material). Briefly: each of the preprocessed microarray datasets was clustered in parallel using Euclidean distance and Hartigan's k-means clustering algorithm. The 'ideal' number of clusters (k) for each dataset was determined within a range of $k = 1$–100 using the jump statistic[53]. Taking the 16 sets of clusters as the input data (Table 1), a weighted co-cluster graph was constructed[16,18]. To select modules, an iterative algorithm was used to extract the sets of probes that are most frequently clustered together in the same datasets, proceeding from the most stringent requirements to the least, as previously described[18]. This iterative process differed from the previous implementation of this algorithm in that the k value was calculated independently for each dataset cluster and the size of the core sub-networks was smaller (10 probes). The algorithm also differed from previous implementations to ensure that the core sub-networks co-clustered in the same datasets. The resulting 382 module set constitutes the third generation of the modular blood transcriptome repertoire constructed since the development of the first generation published in 2008[18], and the second generation published in 2013[17]. Module identifiers (Mxx.xx) were attributed, with the first number indicating the round of selection (the smaller the number the higher the number of datasets in which co-clustering was observed; for M1 it would be 16/16; for M2 it would be 15/16 etc.); the next number represents the order in which it was selected (the smaller the number, the larger the size of the initial seed).

**Module annotation**

*Gene ontology/pathway enrichment*. Module gene lists were investigated using "Database for Annotation Visualization and Integrated Discovery" (DAVID) version 6.7[25]. This database uses a modified Fisher exact test to identify specific

biological/functional categories that are over-represented in gene sets in comparison with a reference set. The top matched DAVID annotation cluster (using default settings), the top matched canonical pathway from the Kyoto Encyclopedia of Genes and Genomes (KEGG), the top matched pathway from BioCarta, and the top matched gene ontology biologic process (GO_BP) and molecular function (GO_MF) terms were identified for each module. Each module was also investigated for significant overlap with two other established blood transcriptome module repertoires[17,23]. The findings are summarized in a module annotation spreadsheet (Supplementary Data 1).

*Gene set annotation (GSAn)*. To further annotate the modules, a new alternative to statistical enrichment analysis tool called GSAn was applied[26]. Statistical enrichment methods may have limitations[36–38], as these methods tend to focus on the subpart of the most studied genes and to provide gene set annotation results that cover a limited number of the well-annotated genes. To address these issues, GSAn offers: (i) an original method that combines semantic similarity measures and data mining approaches to achieve a unified and synthetic annotation of a gene set of interest, and (ii) a visualization approach that facilitates interactive exploration of the gene set annotation results according to the hierarchical structure of gene ontology[13]. The tool is available online: https://gsan.labri.fr/. A page listing analysis results for all 382 generation 3 blood transcriptome modules can be accessed at: https://ayllonbe.github.io/modulesV3/index.html.

*Pathway enrichment analyses*. Ingenuity Pathway Analysis was applied to determine pathway enrichment for each module (Qiagen, Valencia, CA,USA).

*Literature profiling*. Literature Lab™ (LitLab; from Acumenta Biotech, Boston, MA) was used to associate genes within a particular module to terms used in PubMed abstracts[54]. Association scores reflecting the strength of the associations were used to calculate the "Product Scores". The top three terms that showed the strongest association and highest "Product Scores" were used to create the functional annotation. A similar approach using LitLab has been previously reported[49]. The steps taken to annotate all 382 modules are summarized here. All statistical analyses were performed using Microsoft Excel (2010) with Visual Basic for Applications (VBA), Linux-based command line in Mac OS, and R statistical software.

To construct a Product Scores Table, all the terms available in LitLab (>80,000) were listed. Then, the genes in each module were submitted as a list to LitLab Editor and validated manually using LitLab's built-in validation tool and/or NCBI Gene (https://www.ncbi.nlm.nih.gov/gene) prior to submission for analysis using all domains available. After completing the analysis, the summary result page was exported to an xls file. Using the UNIX command line, the exported files were then converted to csv files with the filename appended in the last column of each row and vertically appended. The "merged" file was used to populate the table that included all the available LitLab terms. The top three terms with the highest Product Scores were selected to represent the module functional annotation and are tabulated in column I of the module annotation table (Supplementary Data 1).

*Identification of transcription factor binding motif over-representation*. The Rcis-Target tool in the R package was used to screen modules for enrichment in transcription factor binding motifs[27]. The analysis results have been uploaded to GitHub (https://motoufiq.github.io/DC_Gen3_Module_Analysis/) and are available as interactive circle packing plots (Prezi: Supplementary Table 2).

**Fingerprint grid plot visualization**. Modules were arranged on a grid based on their similarities in patterns of activity across the 16 input datasets, each of them corresponding to a different pathological or physiological state. First, the modules were partitioned using k-means clustering, which generated 38 clusters. Given the possibility of collapsing values of the modules constituting each cluster in a single "aggregate" value, the term "module aggregate" was used to designate each cluster (A1 to A38). Of these 38 k-means clusters, 27 comprised >1 module. The modules were then arranged on a grid with each row corresponding to modules belonging to the same aggregate (Fig. 4). The total number of rows on the grid equaled 27 and the number of columns equaled the largest number of modules for a given aggregate (42 for aggregate A2). For each module, the highest of the two values indicating an increase or a decrease was selected for visualization (e.g. if % increase > % decrease, then a red sport representing % increase is shown). In addition to the Mxx.xx identifiers described above the modules are assigned an identifier that corresponds to their position on the grid: Axx.xx identifiers, with Axx indicating the aggregate (row) number and.xx the order (column). These identifiers are provided in Supplementary Data 1 and in the plots generated via the BloodGen3 R Shiny applications.

**Module fingerprint analysis and visualization**. Downstream analyses and visualizations are supported by the BloodGen3Module R package that we have developed and described in a separate publication[43]. Briefly, the modular analysis was performed using 14,168 probes comprised in the BloodGen3 repertoire. Fold change in expression was computed using gene expression data prior to $\log_2$ transformation. For group comparisons, paired *t*-tests were performed on $\log_2$-transformed data (fold change (FC) cut-off = 1.5; FDR cut-off = 0.1). For

individual-patient analysis, each sample was compared to the mean of control samples in each dataset. Cut-offs were defined by an absolute FC > 1.5 and a difference in gene expression level >10. A module was considered to be "responsive" when the proportion of differentially expressed transcripts (as defined above) was greater than 15%. Data visualization was performed using "ComplexHeatmap"[55].

**Generation of circular plots**. Circular plots were generated to represent module expression at the gene level (Fig. 7). These plots show the fold changes in expression in the case versus control groups in each dataset. Probes that confirmed to the FDR < 0.1 and FC > 1.5 criteria were presented in graded intensity red reflecting the fold change in abundance and those with FC < −1.5 were presented in graded intensity blue. The position of the genes on each of the circular plots is fixed.

**Statistical analyses**. Numerical data were processed and analyzed using R statistical software (version1.1.463-©2009-2018 RStudio, Inc). Student's $t$ test was used to evaluate the significance of differences between groups. $P < 0.05$ was considered to indicate statistical significance, with adjustment by multiple-testing correction when needed (FDR, Benjamini–Hochberg procedure).

**Reporting summary**. Further information on research design is available in the Nature Research Reporting Summary linked to this article.

## Data availability

Transcriptome profiling data that support the findings of this study have been deposited in the NCBI Gene Expression Omnibus (GEO) with the accession code GSE100150.

## Code availability

A R package was developed that supports module repertoire analyses and fingerprint visualizations described in the manuscript. The package is available on GitHub and Bioconductor[56] and described in detail in a separate publication[43] https://github.com/Drinchai/BloodGen3Module https://bioconductor.org/packages/release/bioc/html/BloodGen3Module.html.

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

## Acknowledgements

The authors would like to thank Quynh-Anh Nguyen, Kimberly O'Brien, Dimitry Popov, Michael Mason, and Cate Speake for technical assistance, and Insight Editing London for assistance in editing the manuscript prior to submission. This project has been funded in part by Federal funds from the National Institutes of Health under contract number U01AI082110. Authors affiliated with Sidra Medicine, a member of the Qatar Foundation for Education, Science and Community Development, are fully supported by institutional funding. RJW is supported by the Wellcome Trust (203135, 104803), NIH (U01AI115940) and the Francis Crick Institute, which receives funding from the Wellcome Trust (FC10218), CR UK (FC10218) and UKRI (FC10218). Study samples from the CHAVI cohort were obtained through the support from the NIH, NIAID, AI067854 (the Center for HIV/AIDS Vaccine Immunology).

## Author contributions

Conceptualization: M.C.A., D.R., N.B., D.C. Data curation: N.B., M.A. Visualization: D.R., M.T., D.C. Analysis and interpretation: M.C.A., D.R., N.B., A.A.B., E.W., M.G., B.S.A.K., M.T., D.C. Resources: M.T., M.A., S.R.P., P.K., A.A.B., F.M., P.T., L.C., N.J.C., J.T.P., G.K., A.O.G., M.B., C.B., R.J.W., C.M.G., M.L., G.L., D.B., R.T., F.K., A.M., O.R., K.P., V.P., J.B. Writing of the first draft: D.C. Funding acquisition: G.K., K.P., V.P., O.R., J.B., D.C. Methodology development: M.C.A., D.R., N.B., E.W., D.C. Writing–review & editing: M.C.A., D.R., N.B., M.T., E.W., M.G., B.S.A.K., M.A., S.R.P., P.K. A.A.B., F.M., P.T., L.C., N.J.C., J.T.P., G.K., A.O.G., M.B., C.B., R.J.W., C.M.G., M.L., G.L., D.B., R.T., F.K., A.M., O.R., K.P., V.P., J.B., D.C. The contributors' roles listed above follow the Contributor Roles Taxonomy (CRediT) managed by The Consortia Advancing Standards in Research Administration Information (CASRAI).

## Competing interests

The authors declare no competing interests.

## Additional information

[1]Systems Immunology, Benaroya Research Institute, Seattle, WA, USA. [2]Division of Allergy and Infectious Diseases, University of Washington, Seattle, WA, USA. [3]Research Branch, Sidra Medicine, Doha, Qatar. [4]Baylor Institute for Immunology Research, Baylor Research Institute, Dallas, TX, USA. [5]Inserm U1219 Bordeaux Population Health Research Center, Bordeaux University, Bordeaux, France. [6]LaBRI, CNRS UMR5800, Bordeaux University, Bordeaux, France. [7]Department of Internal Medicine, Hopital Européen, Marseille, France. [8]Aix-Marseille University, C2VN, INSERM 1263, INRA 1260 Marseille, France. [9]Laboratory of Immunoregulation and Infection, The Francis Crick Institute, London, UK. [10]National Heart and Lung Institute, Imperial College London, London, UK. [11]Royal Cornwall Hospitals NHS Trust, Truro, UK. [12]The Francis Crick Institute, London, UK. [13]Department of Infectious Disease, Imperial College, London, UK. [14]Wellcome Center for Infectious Diseases Research in Africa and Department of Medicine, Institute of Infectious Diseases and Molecular Medicine, University of Cape Town Observatory, 7925 Cape Town, Republic of South Africa. [15]UCL Respiratory, Division of Medicine, University College London, London, UK. [16]Centre for Research and Development of Medical Diagnostic Laboratories, Faculty of Associated Medical Sciences, Khon Kaen University, Khon Kaen, Thailand. [17]Baylor College of Medicine & Center for Translational Research on Inflammatory Diseases, Michael E. DeBakey VAMC, Houston, TX, USA. [18]Abigail Wexner Research Institute at Nationwide Children's Hospital and the Ohio State University School of Medicine, Columbus, OH, USA. [19]The Jackson Laboratory for Genomic Medicine, Farmington, CT, USA. [20]Weill Cornell Medicine, New York, NY, USA. [21]These authors contributed equally: Matthew C. Altman, Darawan Rinchai. ✉email: maltman@benaroyaresearch.org; drinchai@sidra.org; dchaussabel@sidra.org

