## [Peer Review File · Nature Communications]

REVIEWER COMMENTS

Reviewer #1 (Remarks to the Author):

Altman et al. constructed the 3rd generation of modular blood transcriptome repertoire using 16 datasets covering a wide range of immune states. Compared to the first 2 generations, the 3rd generation includes more reference datasets, and its module construction algorithm is also improved. In addition to describing the repertoire, the manuscript also demonstrates the utility of the repertoire in great details.

General and specific comments:

1. Altman et al. used in-house generated microarray data and co-clustering network to construct gene modules which were subsequently used to investigate the up- or down- regulation of gene modules in diseases compared to healthy. As mentioned in the discussion section, another set of gene modules were derived from a collection of 2753 public datasets (reference #27). What are the pro and cons of these two repertoires? A more specific comparison between the existing repertoire and the proposed will be informative. In addition, there are also a number of other gene set based analysis such as gene set enrichment analysis (GSEA) or Gene Co-expression Network Analysis (GCNA). How does the method presented in the manuscript compare to those?
2. I believe that this modular blood transcriptome repertoire will be useful to many immunology researchers. Maybe I have missed it; this modular blood transcriptome repertoire doesn't seem to have a name. A short name would be helpful for the readers and users.
3. This 3rd generation of modular blood transcriptome repertoire was built using 16 datasets. If certain datasets are removed or new datasets are added, will the repertoire be dramatically changed? If the reference datasets change, do the modules need to be rebuilt and re-annotated? Will it be done automatically or manually?
4. The presented gene modules are built for blood transcriptome and immune-related signature. Can the framework be applied to build gene modules for other cell types or states, for example cancer? If such framework can be automatically applied to other reference datasets and build the gene modules of interest, it will be very useful.
5. I have tried to read carefully both the main text and supplementary material; it is not clear to me how the densely connected sub-networks are selected (Figure 1). How does the algorithm divide the global co-clustering network into subnetworks?
6. The manuscript mentioned that gene module annotation was done using multiple tools GSA, Literature Lab, IPA, DAVID, Biocarta and etc. How were the results from different tools merged into a consensus annotation as shown in Figure 3A right panel? Was this done automatically by algorithms or via manual curation by scientists?
7. In Figure 2A, what are represented by the rows (one row per dataset or per individual)? What does the color represent (a color legend is needed here)? How the 382 modules were clustered into 38 aggregates?
8. In the module fingerprint grids (for example Figure 2C), how was the order of modules within the aggregate decided?
9. It would be nice to demonstrate the utility on a dataset that was not used as reference to build the modules. Again, I might have missed it.
10. Line 237, "six module" should be "four module"?
11. Line 245, "between these two diseases", not clear which two diseases?
12. Figure 7A does not support the statement at line 324-326.
13. Line 457, Figure 8B should be 9B? This paragraph talks about sepsis but without referencing to the figure.

Reviewer #2 (Remarks to the Author):

The manuscript entitled "Development of a fixed repertoire of blood transcriptome modules based on co-expression patterns across immunological States" by Altman et al., presents a framework for the rapid analysis of blood transcriptomic data based on the co-expression of genes across 16 available datasets regrouping a total of 985 individuals. This approach extends previous work from the same authors, based on older micro-arrays and a lower number of samples.

Briefly, the authors infer co-expression relationships that are robust across all 16 datasets based on illumina HT12 beadchip and define modules of co-expressed genes by searching for densely connected subnetworks in the co-expression graph. Modules with similar expression profiles are then grouped into aggregates to ease visualization. Each module and aggregate is then annotated based on functional enrichment analyses and comparison with reference transcriptome dataset. Functions are also provided to compare the expression of all modules from an aggregate across cohorts, and across individuals from a single cohort. They further apply this approach to study the expression of interferon stimulated genes across immune diseases and individuals.

Overall the manuscript is clearly written and the statistically analyses are sound. Yet, in the current form of the manuscript, my general feeling is that the authors present a tool that may prove useful for a trained user in a HT12-based clinical setting, but has not been designed for a wide usage by the scientific community. I highlight below some points that need to be addressed to improve the manuscript and its reach.

Major comments:

- In the current form of the manuscript, access to module annotation, module visualization and integration with new data is made through 3 different tools (collection of prezi links, shiny app, R package). I think that a fully integrated website, would be preferable and more user friendly. This website could allow the user to upload their own data and visualize their fingerprint along side those of the reference datasets. It could also be accompanied with an R package from automatic queries. At minima, I would expect the prezi annotations link to be easily accessible directly from the shiny app website.

- Usage of the R package did not seem entirely straightforward to me (despite being fluent in R). For instance, Individualcomparison and Groupcomparison functions didn't work unless setting a sample_info argument that is not present in the documentation. I have also been unable to run the fingerprint plot function. User friendliness should be improved by providing examples in the documentation, setting relevant defaults values and implementing informative error messages when an argument is missing or outside the range of expected values.

- The authors chose to define their modules based on an illumina beadchip rather than using RNA-sequencing, despite the latter now being mature (~10 year old) and considered state of the art. This makes me wonder the extent to which the proposed tool can be used for data generated from RNA-sequencing. In theory it shouldn't be a problem, yet a detailed look at the modules identified revealed that the analysis were performed at the probe level rather than the gene level, with some genes being part of multiple clusters (eg. SP100 belonging to M13.17 and M15.64). While I understand that the choice was based on data availability, I think analysis should be made at the gene level to give equal weight to every gene and ensure compatibility with RNA-seq technology.

- The approach described in the paper does not seem to take into account blood cell composition. Yet changes in blood cell composition can lead to coordinated changes in transcript levels, and create co-expression modules with strong functional enrichments. For instance, p15, I327-330, the authors report a dichotomy between disease presenting strong suppression of myeloid response (A34-38) and increased lymphocytic response (A1-A8) and other diseases presenting the opposite pattern. While the author state that the factor driving this dichotomy are unclear, a simple explanation to such a dichotomy would be a difference in the relative % of myeloid to lymphoid cell types in the bloods of the patients. Methods have been developed to infer blood cell composition from microarray/RNAseq data (eg. CIBERSORT) and the authors could use such methods to infer % of the major blood cell types, report these percentages in complement to their analyses, and possibly offer the possibility to compute logFold changes adjusted for cell type proportions.

- Module annotations could be greatly improved by adding information on enrichment of the different modules in genes that are bound by specific Transcription factors, based on databases/softwares such as RcisTarget, or TFEA.chip

Minor comments :

- Numbering of modules (MXX.XX) is rather counterintuitive as it does not align with positions into the transcriptional fingerprint. I would advise to drop the MXX.XX numbering altogether and replace it by AXX.XX where the first number corresponds to the aggregate number (row) and the second to the column in the fingerprint plot.

- When commenting Figure 7A (p15). The authors state that "the first order of separation grouped acute HIV infection, MS, juvenile dermatomyositis and COPD in one cluster" and "the remaining 14 states grouped into another cluster". Yet, when looking at figure 7A, the hierarchical clustering tells a different story with HIV and MS, being separated from all other conditions. Either the figure or the text should be updated in order to ensure that the two are concordant.

Reviewer #3 (Remarks to the Author):

This is a very interesting manuscript and the approach is well designed; the process lends itself to multiple applications and, as you project, it could even include RNAseq analysis. The basis is fold-change over the appropriate control data, so it focuses the analyses on the expertise for each of the 16 illnesses used in the study. It seems to me that there could be more included from the healthy (?) 'controls' from the adult populations, if not of the pediatric controls.

1. In 'methods', you describe the procedures of the collection tube, extraction, etc. Yet it seems that some of the data were generated at different times, based on the references cited and that Illumina chips ver 3 & 4 were used. Could you clarify? Since you have used the transcriptome fold change for the various illnesses, it should not matter that the data were generated at different times and laboratories (the description in methods suggested otherwise). That is your argument for the future RNAseq data.

2. When you examined individuals within an illness group, what did you use as the control? You state that each dataset had matched controls-on what features were they matched? Could it be that some of the alterations you saw in individuals were influenced by whatever control you used or did you compare each individual with the same 'overall' control data for that illness? Did clinical notes about a person help to clarify the differences? In the sepsis group one would expect differences based on the stage of illness at the time of the sample, efficacy of treatment, etc.

3. The web sites are very interesting and certain figures summarize the information; please try to improve the written summaries and major conclusions from those sites.

4. From my download at NATURE, I did not find Supplemental File 1. It would have been quite reassuring to have been able to read it. (Supplemental figures were readily available)

5. It is a complex system and you should make a major effort to more simply explain the steps involved in selection of the clusters and modules. It appeared that some of what I am asking might have been in Supplemental File 1-3.

6. Several places I saw "data is". It is a plural word and the phrase should be 'data are...' lines 538, 539, 851 and perhaps in other sentences. Many sections of the manuscript had the phrase written correctly.

This is a very interesting manuscript and the approach is well designed; the process lends itself to multiple applications and, as you project, it could even include RNAseq analysis. The basis is fold-change over the appropriate control data, so it focuses the analyses on the expertise for each of the 16 illnesses used in the study. It seems to me that there could be more included from the healthy (?) 'controls' from the adult populations, if not of the pediatric controls.

1. In 'methods', you describe the procedures of the collection tube, extraction, etc. Yet it seems that some of the data were generated at different times, based on the references cited and that Illumina chips ver 3 & 4 were used. Could you clarify? Since you have used the transcriptome fold change for the various illnesses, it should not matter that the data were generated at different times and laboratories (the description in methods suggested otherwise). That is your argument for the future RNAseq data.

2. When you examined individuals within an illness group, what did you use as the control? You state that each dataset had matched controls-on what features were they matched? Could it be that some of the alterations you saw in individuals were influenced by whatever control you used or did you compare each individual with the same 'overall' control data for that illness? Did clinical notes about a person help to clarify the differences? In the sepsis group one would expect

differences based on the stage of illness at the time of the sample, efficacy of treatment, etc.

3. The web sites are very interesting and certain figures summarize the information; please try to improve the written summaries and major conclusions from those sites.

4. From my download at NATURE, I did not find Supplemental File 1. It would have been quite reassuring to have been able to read it. (Supplemental figures were readily available)

5. It is a complex system and you should make a major effort to more simply explain the steps involved in selection of the clusters and modules. It appeared that some of what I am asking might have been in Supplemental File 1-3.

6. Several places I saw "data is". It is a plural word and the phrase should be 'data are...' lines 538, 539, 851 and perhaps in other sentences. Many sections of the manuscript had the phrase written correctly.

REVIEWER COMMENTS

Reviewer #1 (Remarks to the Author):

Altman et al. constructed the 3rd generation of modular blood transcriptome repertoire using 16 datasets covering a wide range of immune states. Compared to the first 2 generations, the 3rd generation includes more reference datasets, and its module construction algorithm is also improved. In addition to describing the repertoire, the manuscript also demonstrates the utility of the repertoire in great details.

General and specific comments:

1. Altman et al. used in-house generated microarray data and co-clustering network to construct gene modules which were subsequently used to investigate the up- or down-regulation of gene modules in diseases compared to healthy. As mentioned in the discussion section, another set of gene modules were derived from a collection of 2753 public datasets (reference #27). What are the pro and cons of these two repertoires? A more specific comparison between the existing repertoire and the proposed will be informative. In addition, there are also a number of other gene set based analysis such as gene set enrichment analysis (GSEA) or Gene Co-expression Network Analysis (GCNA). How does the method presented in the manuscript compare to those?

Thank you for this comment. It is indeed useful to provide readers with additional details pertaining to the work of Zhou and Altman, which we have now included in the revised manuscript:

Lines 598-609: "More recently, Zhou and Altman described the development of another fixed transcriptional module repertoire based on the assembly of a collection of 2,753 public datasets that were deposited in the NCBI GEO and encompassed 97,049 unique transcriptome profiles (10). Independent component analysis was then used to resolve co-expression relationships, which led to the identification of a set of 139 transcriptional modules. Subsequently, the biological relevance of these sets and advantages of their use for improving robustness of analyses, especially for smaller datasets, were presented. Overall, this provided additional arguments in favor of the reuse of fixed transcriptional module repertoires, an approach that, to date, has not seen widespread use. However, Zhou and Altman chose datasets generated from a wide range of sample types as input data and the resulting framework may not capture some of the specificities that exist within more narrowly defined biological systems (for example a tissue or cell type such as the blood transcriptome in the case of the BloodGen3 repertoire).

2. I believe that this modular blood transcriptome repertoire will be useful to many immunology researchers. Maybe I have missed it; this modular blood transcriptome repertoire doesn't seem to have a name. A short name would be helpful for the readers and users.

Thank you for your excellent suggestion. We have named this latest blood transcriptome repertoire "BloodGen3". It is now mentioned in the title and throughout the revised manuscript. It is also used in the designation of the R package (BloodGen3Module) and web application (BloodGen3Module App). This is definitely helping not only in bringing all these resources together, but also in describing them in this article.

3. This 3rd generation of modular blood transcriptome repertoire was built using 16 datasets. If certain datasets are removed or new datasets are added, will the repertoire be

dramatically changed? If the reference datasets change, do the modules need to be rebuilt and re-annotated? Will it be done automatically or manually?

The module repertoire could indeed be altered by addition or removal of datasets. However, we do not expect many changes at the least granular “module aggregate level”. Indeed, the dominant functional themes found in BloodGen3 aggregates (plasma cells, interferon, inflammation, cytotoxic/T-cell responses, erythrocytes etc.) were already present in both Gen1 and Gen2. More variations would be expected at the more granular module level. We have previously found these inter-module differences to be biologically meaningful (and conserved from Gen2 to Gen3 in the case of interferon) [Chiche et al. *Arthritis Rheumatol.* 2014 Jun;66(6):1583-95]. This is now mentioned among limitations in the discussion: Lines 663-669.

Building a new repertoire of modules based on a different collection of datasets is not difficult and would not take very long. The development of a companion “interpretation framework” (i.e. including functional annotations, reference profiles and fingerprints) would on the other hand be a rather large undertaking.

Indeed, as mentioned in a follow-on point below, functional annotation, for instance, is largely a manual process at this point. It may be possible to automate it in the future, but resolving functional associations (i.e., “connecting the dots”), will remain predominantly a manual process and improve over time as more analyses/illustrative cases are generated.

4. The presented gene modules are built for blood transcriptome and immune-related signature. Can the framework be applied to build gene modules for other cell types or states, for example cancer? If such framework can be automatically applied to other reference datasets and build the gene modules of interest, it will be very useful.

Thank you for this comment. The approach should indeed be generalizable, and fixed module repertoires could be developed for other systems (such as tumor transcriptomes). We have already developed module repertoires for in vitro systems and specific cell populations [Alsina et al. *Nat Immunol.* 2014 Dec;15(12):1134-42; Banchereau et al. *Nat Commun.* 2014 Oct 22;5:5283.]

The reviewer also touches on a bottleneck/rate limiting step (above and also in point #6 below), which is the development of the annotation framework. However, such a framework would not necessarily have to be as dense as the one we have established here for the BloodGen3 repertoire to be useful.

Some of the supporting resources such as the R package and web applications would need to be customized to accommodate visualization of other repertoires but the effort required should be relatively modest.

5. I have tried to read carefully both the main text and supplementary material; it is not clear to me how the densely connected sub-networks are selected (Figure 1). How does the algorithm divide the global co-clustering network into subnetworks?

Supplementary File 1, which clarified this point, might not have been available for review and we apologize for this. This supplementary method describes the selection process in more detail, as pseudocode, and also as a narrative:

“...At this point, the goal is to extract sets of probes that are most frequently clustered together in the same datasets, proceeding from the most stringent requirements to the least. To accomplish this, an iterative algorithm was used. First, the maximum clique threshold was initialized to the number of input cluster sets, the paraclique threshold (pt) was calculated, and a minimum seed size was chosen (we used 15). The outer loop was begun by creating an unweighted graph through the application of the maximum clique threshold (mct) to the weighted co-cluster graph such that a probe pair, or edge, was connected in the unweighted graph only if the corresponding weight in the co-cluster graph equalled or exceeded this threshold.”

Please let us know if further information is required in addition to the description (and pseudocode) in supplemental File 1.

6. The manuscript mentioned that gene module annotation was done using multiple tools GSA, Literature Lab, IPA, DAVID, Biocarta and etc. How were the results from different tools merged into a consensus annotation as shown in Figure 3A right panel? Was this done automatically by algorithms or via manual curation by scientists?

This step was carried out via manual curation by scientists (and in particular thanks to the assiduous work of one of the junior members of our group, Mr. Mohammed Toufiq, who volunteered to drive the annotation process). He has now added two new elements (“circles”) to the annotation framework: the first shows transcription factor motif enrichment analysis results (using the RcisTarget R package) and the second shows heatmaps from another dataset profiling transcript abundance across an expanded set of leukocyte populations (from Monaco et al. Cell Rep. 2019 Feb 5;26(6):1627-1640.e7.)

7. In Figure 2A, what are represented by the rows (one row per dataset or per individual)? What does the color represent (a color legend is needed here)? How the 382 modules were clustered into 38 aggregates?

Figure 2A shows how modules are grouped into aggregates: the second tier of dimension reduction implemented for the first time with this third generation of modules. We have made changes to the figure and legend in order to improve clarity.

Lines 769-776: Rows on this heatmap correspond each to changes in transcript abundance for a given dataset and for a given direction (i.e. increase or decrease in transcript abundance). These values, the percentages of constitutive probes either increased or decreased within a module, are computed for the 16 datasets used as input for module construction (Table 1). Therefore, in total 32 rows are displayed on this heatmap. Columns correspond to modules comprising the BloodGen3 repertoire ($N=382$). The colors are only associated with module aggregate ID and only serve to illustrate the strategy that was employed for organization of modules on the fingerprint grid plot.

We have tried generating a version of this plot without using colors but we felt the information did not come across as well, therefore reverted to the original.

8. In the module fingerprint grids (for example Figure 2C), how was the order of modules within the aggregate decided?

For earlier repertoires (Gen1 & Gen2), the order in which modules were selected determined their order on a given row of the grid (first to last, from left to right). For the BloodGen3 repertoire, a different strategy was employed to improve readability and facilitate interpretation. Thus, modules within a given aggregate (=row) were grouped first according to

functional annotations and then among the different annotations, from left to right according to alphabetical order. The modules without annotations were added last and ordered in ascending order according to their module ID (i.e. based on the selection process).

This explanation has been added to the text to improve clarity: Lines 635-640.

9. It would be nice to demonstrate the utility on a dataset that was not used as reference to build the modules. Again, I might have missed it.

This is an excellent point, especially given the emphasis on re-usability of such repertoires. The utility has been shown for the earlier iterations (Gen1 and Gen2) and we have now published three proof-of-principle papers using BloodGen3 [Rinchai et al. Clin Transl Med. 2020 Dec;10(8):e244.; Rawat et al. Front Immunol. 2020 Nov 24;11:587946; Rinchai et al. J Transl Med. 2020 Jul 31;18(1):291.]. A new section under results has now been added at the end of the results section that describes these illustrative use cases along with other resources that have also been made available to help support BloodGen3 module repertoire analyses (Lines 546-586: “Development and availability of ancillary resources”).

10. Line 237, “six module” should be “four module”?

Thank you for pointing this out, we have edited this sentence as follows:

Lines 297-298: *We generated fingerprint grid plots for each of the 16 diseases or physiological states (Supplementary file 3); seven of them are shown in Figure 3 as an illustration.*

11. Line 245, “between these two diseases”, not clear which two diseases?

Thank you for your comment. This has been clarified in the revised manuscript by moving the line in question directly after a statement describing changes observed in patients with Melanoma and COPD. (Lines 304-306).

12. Figure 7A does not support the statement at line 324-326.

Thank you for your comment. This point has been addressed in the revised manuscript:

Our statement lines 397-398 of the revised document now reads:

“In the first order of separation, patients with acute HIV infection were grouped in one cluster, while the remaining 14 states were grouped into a second cluster.”

13. Line 457, Figure 8B should be 9B? This paragraph talks about sepsis but without referencing to the figure.

Thank you for your comment, it should have indeed read Figure 9B. The manuscript has been edited accordingly.

Reviewer #2 (Remarks to the Author):

The manuscript entitled “Development of a fixed repertoire of blood transcriptome modules based on co-expression patterns across immunological States” by Altman et al., presents a framework for the rapid analysis of blood transcriptomic data based on the co-expression of genes across 16 available datasets regrouping a total of 985 individuals. This approach extends previous work from the same authors, based on older micro-arrays and a lower number of samples.

Briefly, the authors infer co-expression relationships that are robust across all 16 datasets based on illumina HT12 beadchip and define modules of co-expressed genes by searching for densely connected subnetworks in the co-expression graph. Modules with similar expression profiles are then grouped into aggregates to ease visualization. Each module and aggregate is then annotated based on functional enrichment analyses and comparison with reference transcriptome dataset. Functions are also provided to compare the expression of all modules from an aggregate across cohorts, and across individuals from a single cohort. They further apply this approach to study the expression of interferon stimulated genes across immune diseases and individuals.

Overall the manuscript is clearly written and the statistically analyses are sound. Yet, in the current form of the manuscript, my general feeling is that the authors present a tool that may prove useful for a trained user in a HT12-based clinical setting, but has not been designed for a wide usage by the scientific community. I highlight below some points that need to be addressed to improve the manuscript and its reach.

Major comments:

- In the current form of the manuscript, access to module annotation, module visualization and integration with new data is made through 3 different tools (collection of prezilinks, shiny app, R package). I think that a fully integrated website, would be preferable and more user friendly. This website could allow the user to upload their own data and visualize their fingerprint along side those of the reference datasets. It could also be accompanied with an R package from automatic queries. At minima, I would expect the prezilinks link to be easily accessible directly from the shiny app website.

Thank you for raising this very pertinent point. According to your comments, we have made substantial changes to enhance the utility/usability of this resource. As suggested, we have integrated access to the interactive circle packing plots (Prezi) directly from the redesigned Web app. We also provide links to the following additional resources: 1) the BloodGen3Module R package (analysis and visualization); 2) project-specific Shiny R applications (RSV, COVID-19); and 3) the GXB data browser (providing access to gene level profiles in the 16 reference datasets). We have also added directly in the web application a description of the approaches employed for module construction and downstream analyses to help make this resource more accessible for users without having to refer back to the paper.
<https://drinchai.shinyapps.io/BloodGen3Module/>

In the revised manuscript, we have included a new section describing these applications under results (titled: “Development and availability of ancillary resources”; Lines 546-586).

We are not yet in a position to provide users with the ability to load and analyze private data. Although this may eventually be made available, we consider that most researchers who wish to use this approach to analyze their own data will opt for the R package as a default (hence most of our efforts these past few weeks have been focused on improving this resource, and it has now just been published: (Rinchai et al. *Bioinformatics*. 2021 Feb 24;btab121). We plan

however to make scripts available and provide a step-by-step guide to support the development of project-specific R Shiny web applications (regrouping public and/or private data).

- Usage of the R package did not seem entirely straightforward to me (despite being fluent in R). For instance, Individualcomparison and Groupcomparison functions didn't work unless setting a sample_info argument that is not present in the documentation. I have also been unable to run the fingerprint plot function. User friendliness should be improved by providing examples in the documentation, setting relevant default values and implementing informative error messages when an argument is missing or outside the range of expected values.

Thank you very much for this feedback. A separate manuscript focusing on the description and implementation of this R package was submitted concomitantly for peer-review. The comments of the reviewers largely reflected yours, and very substantial efforts have been made to improve the documentation and general user-friendliness of the BloodGen3module R package. New functions have also been added: <https://github.com/Drinchai/BloodGen3Module>. As indicated in our response to the point above, this paper has just been accepted in the journal "Bioinformatics" (Rinchai et al. Bioinformatics. 2021 Feb 24;btab121). Notably, the package was also submitted to the Bioconductor project with further peer feedback also resulting in further improvements.

- The authors chose to define their modules based on an illumina beadchip rather than using RNA-sequencing, despite the latter now being mature (~10 year old) and considered state of the art. This makes me wonder the extent to which the proposed tool can be used for data generated from RNA-sequencing. In theory it shouldn't be a problem, yet a detailed look at the modules identified revealed that the analysis were performed at the probe level rather than the gene level, with some genes being part of multiple clusters (eg. SP100 belonging to M13.17 and M15.64). While I understand that the choice was based on data availability, I think analysis should be made at the gene level to give equal weight to every gene and ensure compatibility with RNA-seq technology.

The BloodGen3 repertoire is indeed now used routinely to perform analyses using RNAseq data, and some illustrative cases have already been published (Rinchai et al: J Transl Med. 2020 Jul 31;18(1):291, Rawat et al. Front Immunol. 2020 Nov 24;11:587946). Notably, the Gen2 repertoire, also based on whole blood and Illumina beadarrays, has been used to analyze profiles generated from PBMCs using a targeted Nanostring panel (Bhardwaj et al: Nat Cancer. 2020 Dec;1(12):1204–17). There are several other examples of cross-platform and cross-sample type use and these have been mentioned in the revised manuscript.

Lines 579-683: "It is also worth noting that the BloodGen3 repertoire is currently routinely used in our laboratory to perform analyses using RNAseq data (some illustrative cases are already available (32,33). In addition, the "Gen2" repertoire, also based on whole blood and Illumina BeadArrays (3), has been used to analyze profiles generated from peripheral blood mononuclear cells using a targeted 700-transcript Nanostring panel (34)."

Regarding the handling of probes distributed across different modules, there are two possible solutions. We have added the following paragraph to the discussion under limitations of the approach.

Lines 683-695: "It is worth drawing the attention of investigators who wish to employ the framework for the analysis of RNA-seq data that the BloodGen3 repertoire was constructed at the probe-level (those present on the Illumina HT12 v3 Beadarrays). This was to account

for the possibility that changes in abundance for transcript variants measured by distinct probes may differ. And indeed, in some cases different probes mapping to the same genes were distributed across several modules (often within the same aggregate). Thus, for those analyzing data generated at the gene level via RNA-seq the choice would be to assign the same value to duplicate genes found in different modules. Another possibility would be to disregard genes with duplicates altogether, which in our hands shows to have very little impact on the results since they concern only a small minority of genes (388 out of 14,168 probes). Both options can be implemented using functions provided within the BloodGen3Module R package (36).”

- The approach described in the paper does not seem to take into account blood cell composition. Yet changes in blood cell composition can lead to coordinated changes in transcript levels, and create co-expression modules with strong functional enrichments. For instance, p15, l327-330, the authors report a dichotomy between disease presenting strong suppression of myeloid response (A34-38) and increased lymphocytic response (A1-A8) and other diseases presenting the opposite pattern. While the authors state that the factor driving this dichotomy are unclear, a simple explanation to such a dichotomy would be a difference in the relative % of myeloid to lymphoid cell types in the bloods of the patients. Methods have been developed to infer blood cell composition from microarray/RNAseq data (eg . CIBERSORT) and the authors could use such methods to infer % of the major blood cell types, report these percentages in complement to their analyses, and possibly offer the possibility to compute logFold changes adjusted for cell type proportions.

Changes in relative cell composition will indeed be in part driving fluctuations in transcript abundance in whole blood. As such, this is therefore captured through co-expression analyses and indeed, a number of annotations & reference leukocyte profiling datasets associate modules with cell types.

These associations have been made earlier using two reference transcriptome datasets for isolated leukocyte populations; however, prompted by your suggestion, we have included in the annotation framework a third dataset generated by Monaco et al. for the implementation of a deconvolution method similar to CIBERSORT, but better adapted to processing blood transcriptional profiling data (as opposed to tumor profiling data). [ABIS: Monaco et al. Cell Rep. 2019 Feb 5;26(6):1627-1640.e7.]: Lines 294-296. Corresponding heatmaps were generated and added to each of the 382 circle packing plots that are accessible through Prezi.

In effect, the use of both deconvolution methods and module repertoires permits the interpretation of blood transcriptome profiling data in terms of changes in transcript abundance. A notable difference being that relying on module repertoires does not require making a priori choices regarding the sets of leukocyte populations that should be deconvoluted from bulk transcriptome profiling data. One of our recently published illustrative cases highlights this aspect in that it describes a dominant circulating erythroid cell signature (A36, A37, A38) with putative immunosuppressive function that was found to be associated disease severity in patients with respiratory syncytial virus infection [Rinchai et al, Clin Transl Med. 2020 Dec;10(8):e244]. This cell population is not profiled via the CIBERSORT or ABIS methods.

We have also added some discussion of the use of deconvolution methods and module repertoires to resolve changes in cell composition: Lines 641-656.

- Module annotations could be greatly improved by adding information on enrichment of the different modules in genes that are bound by specific Transcription factors, based on databases/software such as RcisTarget, or TFEA.chip

Thank you for this excellent suggestion. The RcisTarget R package was used for this and links to the results were added to the circle packing plot; all the results are available via GitHub, including interactive networks (https://motoufiq.github.io/DC_Gen3_Module_Analysis/).

In the revised manuscript, this has also been mentioned in the text and described in the method section.

Lines 1085-1089: Identification of transcription factor binding motif over-representation: The RcisTarget tool in the R package was used to screen modules for enrichment in transcription factor binding motifs (14). The analysis results have been uploaded to GitHub (https://motoufiq.github.io/DC_Gen3_Module_Analysis/) and are available as interactive circle packing plots (Prezi: Table 3).

Minor comments:

- Numbering of modules (MXX.XX) is rather counterintuitive as it does not align with positions into the transcriptional fingerprint. I would advise to drop the MXX.XX numbering altogether and replace it by AXX.XX where the first number corresponds to the aggregate number (row) and the second to the column in the fingerprint plot.

Thank you for this suggestion. We have updated our R package so that AXX.XX IDs can be used in addition to, or instead of, the MXX.XX IDs, which has been the more traditional denomination. AXX.XX IDs have also been added to the plots generated using the Shiny R application. Numbering has been added to the Prezi landing pages to indicate the position of the modules on the grid.

However, it would be difficult at this point to replace entirely the existing nomenclature throughout the annotation framework as this would be the massive undertaking (i.e. throughout the thousands of analysis reports and heatmaps available via the Prezi circle packing plots). In addition, several illustrative cases employing the pre-existing nomenclature are already published.

It is also worthwhile noting that the original module identifiers (Mxx.xx) do carry some meaningful information, with the first number indicating the round of selection (the smaller the number the higher the number of datasets in which co-clustering was observed; for M1 it would be 16/16; for M2 it would be 15/16 etc.); the next number represents the order in which it was selected (the smaller the number, the larger the size of the initial seed). Granted this is indeed not particularly intuitive.

Relevant written statements were added to the methods section:

Lines 1033-1037: “Module identifiers (Mxx.xx) were attributed, with the first number indicating the round of selection (the smaller the number the higher the number of datasets in which co-clustering was observed; for M1 it would be 16/16; for M2 it would be 15/16 etc.); the next number represents the order in which it was selected (the smaller the number, the larger the size of the initial seed).”

Lines 1102-1106: “In addition to the Mxx.xx identifiers described above the modules are assigned an identifier that corresponds to their position on the grid: Axx.xx identifiers, with Axx indicating the aggregate (row) number and .xx the order (column). These identifiers are

provided in Supplementary File 2 and in the plots generated via the BloodGen3 Shiny R applications.”

- When commenting Figure 7A (p15). The authors state that “the first order of separation grouped acute HIV infection, MS, juvenile dermatomyositis and COPD in one cluster” and “the remaining 14 states grouped into another cluster”. Yet, when looking at figure 7A, the hierarchical clustering tells a different story with HIV and MS, being separated from all other conditions. Either the figure or the text should be updated in order to ensure that the two are concordant.

Thank you for pointing out the discrepancy. The text has now been amended to accurately describe this Figure.

Lines 397-399: *“In the first order of separation, patients with acute HIV infection were grouped in one cluster, while the remaining 14 states were grouped into a second cluster.”*

Reviewer #3 (Remarks to the Author):

This is a very interesting manuscript and the approach is well designed; the process lends itself to multiple applications and, as you project, it could even include RNAseq analysis. The basis is fold-change over the appropriate control data, so it focuses the analyses on the expertise for each of the 16 illnesses used in the study. It seems to me that there could be more included from the healthy (?) ‘controls’ from the adult populations, if not of the pediatric controls.

Thank you for the positive feedback. We have revised the manuscript to clarify some points regarding selection and use of control subjects as outlined below:

1. In ‘methods’, you describe the procedures of the collection tube, extraction, etc. Yet it seems that some of the data were generated at different times, based on the references cited and that Illumina chips ver 3 & 4 were used. Could you clarify? Since you have used the transcriptome fold change for the various illnesses, it should not matter that the data were generated at different times and laboratories (the description in methods suggested otherwise). That is your argument for the future RNAseq data.

Thank you for pointing this out. All the data were in fact generated using the Human HT12 v3 Bead Arrays. This has now been corrected throughout the manuscript.

In addition, our approach is indeed a priori compatible with reuse of publicly available data, which was probably less common when we started this line of work (approximately 2005 for generation 1). We were also fortunate enough to have access for the BloodGen3 construction to a collection of well-characterized blood transcriptome datasets generated at the same facility and obtained across different projects using harmonized protocols. This point was clarified in the manuscript:

Lines 671-675: *“Our choice to use this technology was based on availability at our institute for the construction of the BloodGen3 repertoire of a collection of well-characterized blood transcriptome datasets generated at the same facility and obtained across different projects using harmonized protocols.”*

2. When you examined individuals within an illness group, what did you use as the control? You state that each dataset had matched controls-on what features were they matched? Could it be that some of the alterations you saw in individuals were influenced by whatever control you used or did you compare each individual with the same 'overall' control data for that illness? Did clinical notes about a person help to clarify the differences? In the sepsis group one would expect differences based on the stage of illness at the time of the sample, efficacy of treatment, etc.

Thank you for highlighting this important point. As is customary, subjects included in the control group were primarily selected based on demographics (age, gender and ethnicity). However, for the adult sepsis dataset, diabetes was also taken into account since it is an important risk factor and is highly prevalent among patients with septicemic melioidosis (North-Eastern Thailand was the setting of this particular study).

We added language in order to clarify this point:

Lines 938-940: The gene expression datasets selected to cover major classes of immune states (Table 2) were required to have at least 25 samples in total, and at least 20% of the total samples were required to be controls matched for gender, age and ethnicity.

Lines 953-955: Factors accounted for in the selection of subjects in the control group included gender, age and type 2 diabetes diagnosis, the latter being a risk factor for septicemic melioidosis.

Furthermore, when performing individual patient level analyses, the entire control group is used as a reference, which should reduce risk of bias that would arise from matching to a single control (Lines 1114-1116).

Finally, it is also worth noting that the module repertoire construction itself was performed independently of sample grouping information. Indeed, only instances of gene-gene co-clustering were recorded and used to inform gene grouping in the modules. Levels of transcript abundance in cases were compared to controls only while performing downstream analyses using the BloodGen3 repertoire, after its construction and characterization.

3. The web sites are very interesting and certain figures summarize the information; please try to improve the written summaries and major conclusions from those sites.

Thank you for your suggestion. In the revised manuscript, we have included a new subsection under results describing these applications in the results section (under the subsection titled: "Development and availability of ancillary resources; Lines 546-584).

4. From my download at NATURE, I did not find Supplemental File 1. It would have been quite reassuring to have been able to read it. (Supplemental figures were readily available)

We apologize for this problem. Supplemental File 1 has been uploaded again and will hopefully now be accessible. This file provides supplemental methods/information about module construction.

5. It is a complex system and you should make a major effort to more simply explain the steps involved in selection of the clusters and modules. It appeared that some of what I am asking might have been in Supplemental File 1-3.

Supplemental File 1 provides a more detailed explanation, which should help clarify this point. In addition, we also added details to the legend of Figure 1 (Lines 663-667).

6. Several places I saw “data is”. It is a plural word and the phrase should be ‘data are...’ lines 538, 539, 851 and perhaps in other sentences. Many sections of the manuscript had the phrase written correctly.

Thank you for pointing out this error. This has now been corrected.

REVIEWERS' COMMENTS

Reviewer #1 (Remarks to the Author):

Altman et al. have addressed all the comments. In addition, the authors have introduced several recently published or accepted studies, which include their R package BloodGen3Module and the applications of BloodGen3Module by others. I have no further comments.

Reviewer #2 (Remarks to the Author):

First, I would like to apologize for the delays in sending this review.

The revised version of the Manuscript entitled « Development of the BloodGen3 module repertoire, a novel framework for the analysis, visualization and interpretation of blood transcriptome data » by Altman et al has been largely improved from the first draft and I feel that most of my previous comments have been appropriately addressed by the authors.

The authors have now included analysis of TF binding sites as suggested, and have extended annotations of cell type + improved the discussion on the importance of cell proportions in driving differences in blood transcriptional profiles.

With the new integrated interface, I believe that the proposed tool has potential value in clinical practice, and I appreciate the effort done to make it more accessible. Yet, I still regret that the associated R package does not seem to provide example datasets for which the authors provide code to run the method with a brief description of the expected results, as is commonly done in R packages, and that the examples are limited to a command with the default parameters.

While I still believe that this repertoire could have benefitted from being defined based on RNA-seq data, I agree with the authors that it is “unlikely that entire co-expressed gene sets would be missed by microarrays to the extent that it would lead to the production of a repertoire that is fundamentally different to that produced using RNA-seq data » I thus agree that this could be a useful resource as it is, and merits publication.

Reviewer #3 (Remarks to the Author):

The analytical package, BloodGen3, offers an interesting process for analysis of massive datasets and has presented a means to contrast and compare host genomic responses to 16 illnesses which involve immune dysfunction.

The responses to the queries from each of the reviewers has been quite complete and the points raised are now included in the manuscript. This process has resulted in a much more understandable presentation. The comparisons of the 16 illnesses using the identification of transcriptomic clusters to compare and contrast the immune responses provides an innovative framework for in-depth analysis and enables similarities and contrasts among the illnesses presented. It is a complex system and the authors have done a masterful job in presenting the overview and potential utility of this approach. It is the process of analysis of these massive datasets which is key and this study well-illustrates the value of the tools and their potential for usefulness throughout the field. They have made good progress in refining the approach and have applied it to other illnesses as referenced in other publications. Although the data in this manuscript were from microarray studies and over many years, their process has enabled comparisons despite the differences in the physical tools used and describe usefulness for RNAseq studies.

REVIEWERS' COMMENTS

Reviewer #1 (Remarks to the Author):

Altman et al. have addressed all the comments. In addition, the authors have introduced several recently published or accepted studies, which include their R package BloodGen3Module and the applications of BloodGen3Module by others. I have not further comments.

>> Thank you for your time. We indeed hope that the use cases we recently published will prove helpful to those who are interested in implementing this approach.

Reviewer #2 (Remarks to the Author):

First, I would like to apologize for the delays in sending this review.

The revised version of the Manuscript entitled « Development of the BloodGen3 module repertoire, a novel framework for the analysis, visualization and interpretation of blood transcriptome data » by Altman et al has been largely improved from the first draft and I feel that most of my previous comments have been appropriately addressed by the authors.

>> Great to hear that earlier comments were satisfactorily addressed. It certainly helped improve the manuscript and clarify several key points.

The authors have now included analysis of TF binding sites as suggested, and have extended annotations of cell type + improved the discussion on the importance of cell proportions in driving differences in blood transcriptional profiles.

With the new integrated interface, I believe that the proposed tool has potential value in clinical practice, and I appreciate the effort done to make it more accessible. Yet, I still regret that the associated R package does not seem to provide example datasets for which the authors provide code to run the method with a brief description of the expected results, as is commonly done in R packages, and that the examples are limited to a command with the default parameters.

>> Indeed, it is important that such a dataset be provided for testing / troubleshooting. We have worked with curators at Bioconductor over the past couple of months to have the BloodGen3Module package included as part of their latest release – and this is now done (as of the end of May). As part of this effort an “experiment package” was created, whereby an example dataset (GSE13015) was parsed into a SummarizedExperiment object - that is now available in ExperimentHub. This should make it very easy for users to test the package and replicate the plots shown in our recent Bioinformatics article (<https://pubmed.ncbi.nlm.nih.gov/33624743/>)

Experiment package:

<https://bioconductor.org/packages/release/data/experiment/html/GSE13015.html>

Software package:

<https://bioconductor.org/packages/release/bioc/html/BloodGen3Module.html>

The data can be retrieved via one of two methods:

```
#1
if (!requireNamespace("BiocManager", quietly = TRUE))
  install.packages("BiocManager")
```

```
BiocManager::install("GSE13015")
```

```
#2
library(ExperimentHub)
dat = ExperimentHub()
hub = query(dat , "GSE13015")
GSE13015_matrix
= hub[["EH5429"]]
```

While I still believe that this repertoire could have benefitted from being defined based on RNA-seq data, I agree with the authors that it is “unlikely that entire co-expressed gene sets would be missed by microarrays to the extent that it would lead to the production of a repertoire that is fundamentally different to that produced using RNA-seq data » I thus agree that this could be useful resource as it is, and merits publication.

>> We have analyzed several new RNAseq datasets over the past few months/weeks and the BloodGen3 repertoire has indeed performed very well. We nonetheless agree that with this approach now established it will be time to turn our attention to RNAseq, to not only develop repertoires for bulk whole blood but also leukocyte populations transcriptomes.

Reviewer #3 (Remarks to the Author):

The analytical package, BloodGen3, offers an interesting process for analysis of massive datasets and has presented a means to contrast and compare host genomic responses to 16 illnesses which involve immune dysfunction.

The responses to the queries from each of the reviewers has been quite complete and the points raised are now included in the manuscript. This process has resulted in a much more understandable presentation. The comparisons of the 16 illnesses using the identification of transcriptomic clusters to compare and contrast the immune responses provides an innovative framework for in-depth analysis and enables similarities and contrasts among the illnesses presented. It is a complex system and the authors have done a masterful job in presenting the overview and potential utility of this approach. It is the process of analysis of these massive datasets which is key and this study well-illustrates the value of the tools and their potential for usefulness throughout the field. They have made good progress in refining the approach and have applied it to other illnesses as referenced in other publications.

Although the data in this manuscript were from microarray studies and over many years, their process has enabled comparisons despite the differences in the physical tools used and describe usefulness for RNAseq studies.

>> Thank you for your encouraging comments! We also appreciate the time and effort spent evaluating our work and the constructive comments that have led to very substantial improvements.